# Residual force enhancement is affected more by quadriceps muscle length than stretch amplitude

Patrick Bakenecker[1]*, Tobias Weingarten[1], Daniel Hahn[1,2]*, Brent Raiteri[1]*

[1]Human Movement Science, Faculty of Sport Science, Ruhr University Bochum, Bochum, Germany; [2]School of Human Movement and Nutrition Sciences, University of Queensland, Brisbane, Australia

**Abstract** Little is known about how muscle length affects residual force enhancement (rFE) in humans. We therefore investigated rFE at short, long, and very long muscle lengths within the human quadriceps and patellar tendon (PT) using conventional dynamometry with motion capture (rFE$_{TQ}$) and a new, non-invasive shear-wave tensiometry technique (rFE$_{WS}$). Eleven healthy male participants performed submaximal (50% max.) EMG-matched fixed-end reference and stretch-hold contractions across these muscle lengths while muscle fascicle length changes of the vastus lateralis (VL) were captured using B-mode ultrasound. We found significant rFE$_{TQ}$ at long (7±5%) and very long (12±8%), but not short (2±5%) muscle lengths, whereas rFE$_{WS}$ was only significant at the very long (38±27%), but not short (8±12%) or long (6±10%) muscle lengths. We also found significant relationships between VL fascicle length and rFE$_{TQ}$ ($r$=0.63, p=0.001) and rFE$_{WS}$ ($r$=0.52, p=0.017), but relationships were not significant between VL fascicle stretch amplitude and rFE$_{TQ}$ ($r$=0.33, p=0.126) or rFE$_{WS}$ ($r$=0.29, p=0.201). Squared PT shear-wave-speed-angle relationships did not agree with estimated PT force-angle relationships, which indicates that estimating PT loads from shear-wave tensiometry might be inaccurate. We conclude that increasing muscle length rather than stretch amplitude contributes more to rFE during submaximal voluntary contractions of the human quadriceps.

**\*For correspondence:**
Patrick.Bakenecker@rub.de (PB);
Daniel.Hahn@rub.de (DH);
Brent.Raiteri@rub.de (BR)

**Competing interest:** The authors declare that no competing interests exist.

## Editor's evaluation

This manuscript will be of interest to those who study human performance and muscle physiology. The study involved a very careful evaluation of a phenomenon whereby eccentric contraction increases the force generating capacity of skeletal muscle. Several modern techniques including measurements of muscle fascicle length and kinematics were used. The results sharpen our understanding of the relationships between muscle length and performance.

## Introduction

Residual force enhancement (rFE) is a history-dependent property of skeletal muscle and is defined as the enhanced isometric (i.e. steady-state) force following active muscle lengthening (i.e. stretch) relative to the isometric force obtained during a fixed-end reference contraction at the same muscle length and level of activation (*Cook and McDonagh, 1995*; *Edman et al., 1978*). rFE has been observed across muscle structural scales ranging from single sarcomeres (*Leonard et al., 2010*) to isolated fibres from animals (*Abbott and Aubert, 1952*) and humans (*Pinnell et al., 2019*) to whole human muscles working in vivo (*Cook and McDonagh, 1995*) during artificially evoked and voluntary contractions (*Lee and Herzog, 2002*; *Seiberl et al., 2015*).

In vitro and in situ experiments have shown that rFE is independent of stretch velocity (*Edman et al., 1978*; *Tilp et al., 2009*), but dependent on muscle length, whereby rFE is greater when the stretch ends at longer muscle lengths (*Bullimore et al., 2007*; *Hisey et al., 2009*). rFE generally increases with increasing stretch amplitude (*Edman et al., 1982*; *Edman et al., 1978*) until some critical amplitude on the descending limb of the force-length relationship (*Bullimore et al., 2007*; *Hisey et al., 2009*), but increasing stretch amplitude does not necessarily increase rFE when the stretch ends at shorter muscle lengths (*Hisey et al., 2009*). In comparison with in vitro experiments, stretch amplitude and muscle length are more difficult to control in vivo because there are multiple muscle-tendon units (MTUs) that cross a joint of interest, and these relatively more compliant MTUs have varying moment arms over a joint's range of motion that change with contraction intensity (*Tsaopoulos et al., 2007*). In a recent meta-analysis, *de Campos et al., 2022* analysed how stretch amplitude affects in vivo rFE and concluded that rFE, which was estimated from measured net joint torques, was not dependent on stretch amplitude (i.e. the amount of joint rotation) during artificially evoked or voluntary contractions. However, this review neglected the previously reported in situ-based interaction between stretch amplitude and muscle length on rFE (*Hisey et al., 2009*), largely because very few in vivo studies have estimated the operating region of an individual muscle of interest in relation to its optimum length for maximum isometric force production.

In vivo experiments that have examined how muscle length affects rFE show somewhat conflicting results. *Power et al., 2013* showed that rFE occurs at both short and long muscle lengths, whereas other studies found rFE only at long muscle lengths (*Bakenecker et al., 2020*; *Fukutani et al., 2017*; *Shim and Garner, 2012*). One potential reason for these conflicting results could be that most studies tested at identical joint angles across participants for the different muscle length conditions, even though this could have resulted in rFE being examined only over the ascending *or* descending limb of the force-length relationship in some participants, and over both ascending *and* descending limbs in other participants (*Bakenecker et al., 2019*). Consequently, it remains unclear how in vivo rFE is affected by muscle length.

Another difficulty with assessing the influence of muscle length on in vivo rFE is that muscle or tendon forces are typically estimated using net joint torque measurements and literature-based moment arms (*Bakenecker et al., 2020*). However, using literature-based moment arms might be inaccurate as moment arms can vary between individuals and genders (*Nisell, 1985*), are affected by contraction intensity (*Arampatzis et al., 2004*; *Tsaopoulos et al., 2007*), and are influenced by the determination method (*Bakenecker et al., 2019*). Secondly, net joint torque measurements to infer rFE can be problematic because torque is influenced by (1) gravitational forces, (2) passive and active forces generated by agonistic and antagonistic muscles, and (3) misalignment between the joint axis of interest and dynamometer axis can occur (*Arampatzis et al., 2004*). These factors might be partly responsible for the discrepancies between in vitro and in vivo findings.

One potential solution to more directly quantify in vivo rFE involves using a new, non-invasive technique known as shear-wave tensiometry, which was introduced by *Martin et al., 2018*. These authors previously showed that the shear-wave propagation velocity in free tendons depends primarily on axial stress under physiological loads, and that in vivo tendon shear-wave speeds can be tracked by a shear-wave tensiometer, which consists of a skin-mounted tapping device and two miniature accelerometers. The squared tendon shear-wave speeds measured with this device were shown to strongly correlate with joint torques during fixed-end knee extension and plantar flexion contractions (*Martin et al., 2018*). Squared Achilles and patellar tendon (PT) shear-wave speeds were also shown to strongly correlate with joint torques that were estimated using inverse dynamics during sections of the gait cycle (*Martin et al., 2018*). The usability of calibrated shear-wave tensiometers to assess in vivo Achilles tendon stress was later confirmed in experiments by the same group (*Keuler et al., 2019*).

Therefore, the aim of this study was to examine whether rFE is muscle length dependent within the human quadriceps, and this was addressed using conventional dynamometry with motion capture and shear-wave tensiometry to correct measured knee extension torques and estimate PT loads, respectively. We tested sixteen participants who performed submaximal voluntary EMG-matched (50% of the knee joint angle-specific maximum superficial quadriceps muscle activity) and rotation-amplitude-matched (15° knee joint rotation) stretch-hold and fixed-end reference contractions at short, long, and very long muscle lengths. Stretch-hold contractions indicate that the quadriceps' MTUs were

actively stretched by a triggered rotation of the dynamometer crank arm before being held at a constant length. Fixed-end contractions indicate that the quadriceps' MTU lengths remained relatively constant for the duration of the contraction. To ensure that rFE was not influenced by differences in the muscles' isometric force capacities at short and long muscle lengths, we attempted to match PT loads at short and long muscle lengths. Based on the findings from in vitro experiments (*Bullimore et al., 2007*; *Hisey et al., 2009*) and the mechanisms predicted to contribute to greater rFE at longer muscle lengths, such as sarcomere length non-uniformities (*Morgan, 1990*) and increased titin forces at longer muscle lengths (*Flann et al., 2011*), we hypothesised that rFE would increase with increasing muscle length rather than increasing stretch amplitude, which was partly verified by imaging muscle fascicle lengths and length changes from one muscle, the vastus lateralis (VL), using B-mode ultrasound. Based on the findings of *Martin et al., 2018*, we also expected a strong agreement (≥90%) between squared PT shear-wave speed and the PT force estimated from the corrected knee extension torque.

## Results

### Exclusions

Five participants did not complete the experiment because of problems with estimating their PT shear-wave speeds. No outliers in EMG, torque, or VL fascicle length or length change data were detected from the remaining 11 participants. However, at the long muscle length, two outliers (4 median absolute deviations [MAD]: 9316 $m^2$ $s^{-2}$ [values: 29,671 and 39,375]) from the stretch-hold condition, and two outliers (4 MAD: 9544 $m^2$ $s^{-2}$ [values: 22,204 and 27,710]) from the fixed-end reference condition were detected in the squared PT shear-wave-speed data. Consequently, the sample sizes for squared shear-wave-speed and $rFE_{WS}$ data for the short, long, and very long muscles lengths were 11, 9, and 11, respectively.

### Muscle activity levels

A two-way repeated-measures ANOVA on the mean time-matched steady-state summed muscle activity level data from the superficial quadriceps' muscles (VL, rectus femoris [RF], and vastus medialis [VM]) found a significant main effect of contraction condition ($F_{1,10}$ = 12.59, p=0.005). However, mean differences between stretch-hold and fixed-end reference contractions were within the allowed variation (<6% difference between the desired and recorded EMG levels) and not significant according to Sidak post hoc comparisons at the short (–1.2% [95% CI: –3.4 to 1.0], p=0.396), long (–1.0% [95% CI: –2.9 to 0.8], p=0.367), or very long muscle lengths (–1.1% [95% CI: –2.2 to 0.1], p=0.076). There was

**Table 1.** Mean values and standard deviations for corrected knee extension torque, squared patellar tendon (PT) shear-wave speed, estimated PT force, normalised EMG amplitudes of the vastus lateralis (VL), vastus medialis (VM), rectus femoris (RF), and their sum, as well as VL fascicle length and fascicle stretch magnitudes during the stretch-hold contractions compared with the time-matched fixed-end reference contractions at the short, long, and very long muscle lengths.

| Muscle length | Short | | Long | | Very long | |
|---|---|---|---|---|---|---|
| Contraction condition | Stretch-hold | Fixed-end | Stretch-hold | Fixed-end | Stretch-hold | Fixed-end |
| Knee extension torque (Nm) | 210.1±48.6 | 207.8±53.1 | 253.7±32.2* | 236.0±26.3 | 210.8±29.0* | 188.9±24.1 |
| PT shear-wave speed $(ms^{-1})^2$ | 3900±2770 | 3677±2703 | 2872±1179 | 2775±1215 | 2601±1554* | 1800±846 |
| Estimated PT force (N) | 4321±1012 | 4272±1108 | 5365±687 | 5078±564 | 4804±774 | 4321±619 |
| VL EMG amplitude (% MVC) | 44.65±5.78 | 45.35±6.50 | 43.12±4.19 | 43.62±5.05 | 43.68±5.70 | 43.38±4.15 |
| VM EMG amplitude (% MVC) | 41.68±5.25 | 44.04±5.95 | 43.80±4.84 | 43.50±4.26 | 42.70±4.28 | 43.59±4.21 |
| RF EMG amplitude (% MVC) | 51.60±10.86 | 52.53±10.60 | 48.86±8.66 | 52.12±7.75 | 45.26±6.98 | 48.44±8.19 |
| Summed EMG amplitude (% MVC) | 46.91±2.35 | 48.08±3.14 | 46.84±2.06 | 47.86±1.87 | 47.17±3.30 | 48.22±2.77 |
| VL fascicle length (mm) | 99±34 | 99±34 | 122±24 | 124±24 | 142±36 | 143±35 |
| VL fascicle stretch (mm) | 3.7±2.1 | | 6.0±1.8 | | 6.3±2.8 | |

*Significantly different to fixed-end reference condition at p<0.05. MVC, maximal voluntary contraction.

no significant main effect of muscle length ($F_{1.53,15.26}$ = 0.05, p=0.914), and no significant interaction between contraction condition and muscle length ($F_{1.82,18.15}$ = 0.01, p=0.982). Mean time-matched steady-state biceps femoris muscle activity levels (*n*=7) during stretch-hold and fixed-end reference contractions were 4.6%±2.9% and 5.0%±3.3%, respectively. Descriptive statistics are reported in *Table 1*.

### Corrected knee extension torques

Mean time-matched steady-state corrected knee extension torques during the stretch-hold and fixed-end reference contractions were 210.1±48.6 and 207.8±53.1 Nm at the short muscle length, 253.7±32.2 and 236.0±26.3 Nm at the long muscle length, and 210.8±29.0 and 188.9±24.1 Nm at the very long muscle length. A two-way repeated-measures ANOVA on corrected torque data revealed significant main effects of contraction condition ($F_{1,10}$ = 34.03, p<0.001) and muscle length ($F_{1.15,11.46}$ = 10.81, p=0.006), and a significant interaction between contraction condition and muscle length ($F_{1.99,19.85}$ = 7.96, p=0.003). Sidak post hoc comparisons revealed no significant mean difference between the stretch-hold and fixed-end reference conditions at the short muscle length (2.4 Nm [95% CI: –5.8 to 10.5], p=0.813), but significant mean differences at the long (17.7 Nm [95% CI: 5.8 to 29.5], p=0.005), and very long muscle lengths (21.9 Nm [95% CI: 9.6 to 34.2], p=0.001).

### Squared PT shear-wave speeds

Mean time-matched steady-state squared PT shear-wave speeds during the respective stretch-hold and fixed-end reference contractions were 3900±2770 and 3677±2703 $m^2\,s^{-2}$ at the short muscle length, 2872±1179 and 2775±1215 $m^2\,s^{-2}$ at the long muscle length, and 2601±1554 and 1800±846 $m^2\,s^{-2}$ at the very long muscle length. A two-way repeated-measures mixed-effects analysis on squared shear-wave-speed data revealed significant main effects of contraction condition ($F_{1,10}$ = 13.18, p=0.005) and muscle length ($F_{1.35,13.52}$ = 5.00, p=0.034), and a significant interaction between contraction condition and muscle length ($F_{1.69,13.51}$ = 4.52, p=0.036). Sidak post hoc comparisons revealed no significant mean differences between the stretch-hold and fixed-end reference conditions at the short (223 $m^2\,s^{-2}$ [95% CI: –192 to 638], p=0.399) or long muscle lengths (97 $m^2\,s^{-2}$ [95% CI: –207 to 402], p=0.745), but a significant mean difference at the very long muscle length (802 $m^2\,s^{-2}$ [95% CI: 130 to 1473], p=0.020). Tukey post hoc comparisons revealed that the mean differences in squared shear-wave speeds were significant during the fixed-end reference contractions between the short and very long (1878 $m^2\,s^{-2}$ [95% CI: 145 to 3610], p=0.034) and long and very long muscle lengths (975 $m^2\,s^{-2}$ [95% CI: 81 to 1868], p=0.034), but not significantly different between the short and long muscle lengths (903 $m^2\,s^{-2}$ [95% CI: –1197 to 3003], p=0.471) or between muscle lengths during the stretch-hold contractions (overall: 270 to 1299 $m^2\,s^{-2}$ [95% CI: –871 to 2834], p≥0.060).

### Residual force enhancement

$rFE_{TQ}$ values of 2±5%, 7±5%, and 12±8% were calculated at the short, long, and very long muscle lengths, respectively. $rFE_{WS}$ values at the same respective lengths were 8±12%, 6±10%, and 38±27%. A two-way repeated-measures mixed-effects analysis on rFE data revealed significant main effects of the rFE determination method ($F_{1,10}$ = 13.88, p=0.004) and muscle length ($F_{1.51,15.07}$ = 11.07, p=0.002), as well as a significant interaction between the rFE determination method and muscle length ($F_{1.39,12.49}$ = 9.43, p=0.006). Sidak post hoc comparisons revealed that rFE mean differences were not significant between corrected torque and shear-wave-speed methods at the short (–6% [95% CI: –15 to 3], p=0.248) or long muscle lengths (2% [95% CI: –10 to 13], p=0.976), but there was a significant difference between $rFE_{TQ}$ and $rFE_{WS}$ at the very long muscle length (–27% [95% CI: –45 to –8], p=0.006). Individual results and descriptive statistics are shown in *Figure 1A–B* and *Table 1*, respectively.

### VL muscle fascicle lengths and length changes

During the stretch-hold contractions, VL muscle fascicles stretched by 3.7±2.1 mm to the short muscle length, 6.0± 1.8 mm to the long muscle length, and 6.3±2.8 mm to the very long muscle length. A one-way repeated-measures ANOVA revealed a significant main effect of muscle length on the magnitude of VL muscle fascicle stretch ($F_{1.86,18.57}$ = 9.10, p=0.002). Tukey post hoc comparisons found significant mean differences in VL fascicle stretch between the short and long (–2.3 [95% CI: –3.9 to –0.7], p=0.007), and short and very long muscle lengths (–2.6 mm [95% CI: –4.7 to –0.6], p=0.013),

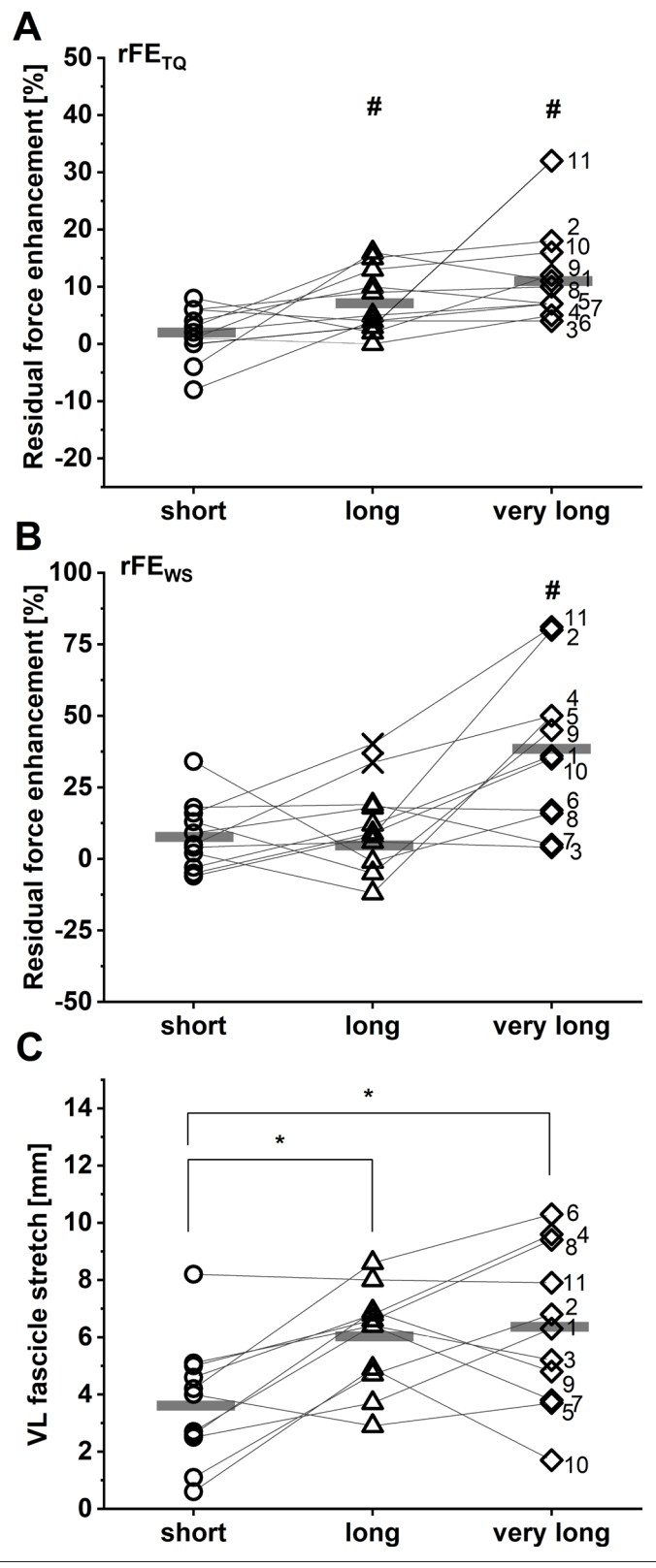

**Figure 1.** Individual and mean residual force enhancement (rFE) magnitudes (*n*=11) based on corrected torque (**A**) and squared patellar tendon (PT) shear-wave-speed recordings (**B**) at the short (dots), long (triangles), and very long (squares) muscle lengths. Individual and mean vastus lateralis (VL) muscle fascicle stretch amplitudes are shown in C (*n*=11). Unfilled symbols represent the individual values and horizontal bars indicate the group mean

*Figure 1 continued on next page*

*Figure 1 continued*

for each condition. Outliers that were excluded from analysis are indicated with a X in B (*n*=2). Grey lines and black numbers distinguish between participants. *Indicates a significant difference between muscle length conditions (p<0.05). #Indicates significant rFE based on corrected knee extension torque (**A**) and shear-wave-speed (**B**) measurements (p<0.05).

but no significant mean difference between the long and very long muscle lengths (–0.3 mm [95% CI: –2.2 to 1.6], p=0.884) (*Figure 1C*). No significant repeated-measures linear relationships were found between VL fascicle stretch amplitude and $rFE_{TQ}$ (*r*=0.33, 95% CI: –0.12 to 0.67, p=0.126) or $rFE_{WS}$ (*r*=0.29, 95% CI: –0.19 to 0.66, p=0.201) (*Figure 2*).

A two-way repeated-measures ANOVA on absolute VL muscle fascicle length data revealed significant main effects of contraction condition ($F_{1,10}$ = 9.72, p=0.011) and fascicle length ($F_{1.98,19.78}$ = 46.56, p<0.001), and a significant interaction between contraction condition and fascicle length ($F_{1.98,19.76}$ = 5.09, p=0.017). Mean differences in fascicle length between stretch-hold and fixed-end reference contractions were significant according to Sidak post hoc comparisons at the long muscle length only (–2 mm [95% CI: –3 to –1], p=0.002), but not at the short (0 mm [95% CI: –1 to 2], p=0.961) or very long muscle lengths (–1 mm [95% CI: –2 to 0], p=0.187). Considering that the mean standard deviation and mean range of fascicle lengths for the fixed-end reference condition at the long muscle length was 2 and 4 mm across participants, respectively, the significant difference in fascicle lengths between the stretch-hold and fixed-end reference contractions at the long muscle length is not likely to be meaningful as it falls within the measurement variability. Tukey post hoc comparisons showed that the

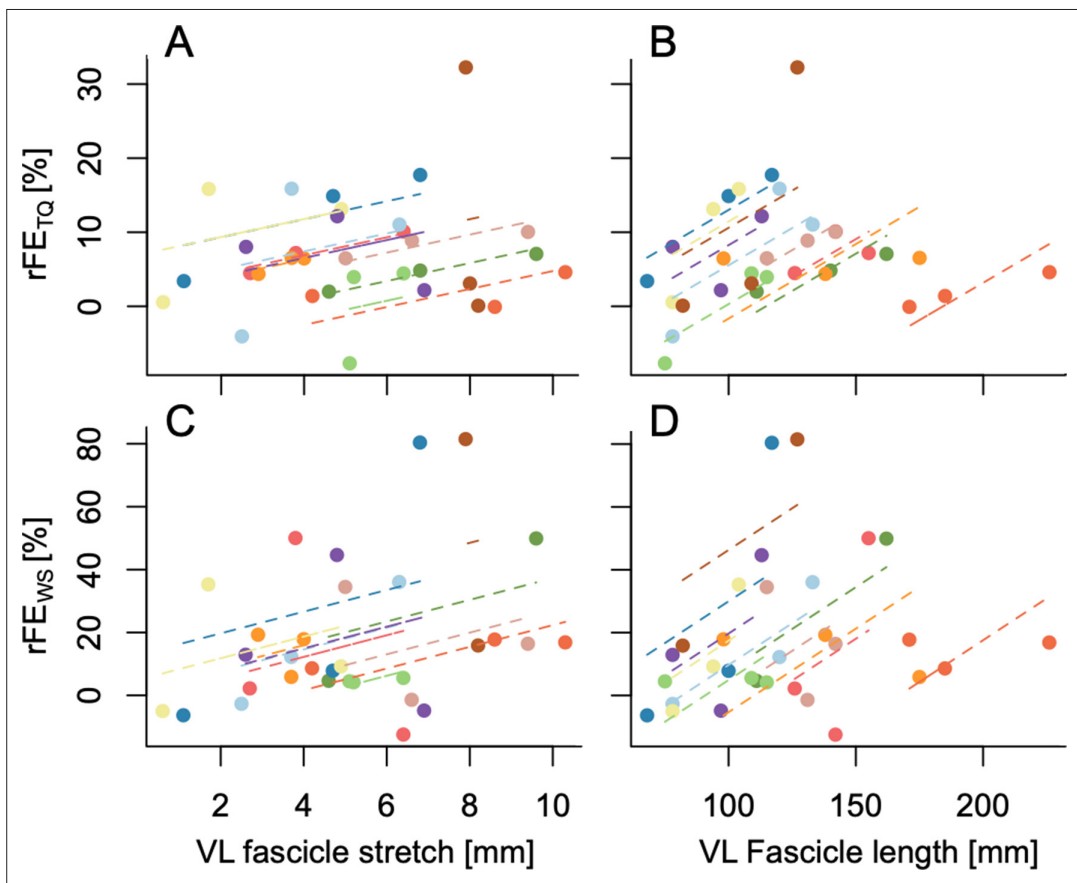

**Figure 2.** Repeated-measures linear relationships between $rFE_{TQ}$ and vastus lateralis (VL) fascicle stretch amplitude (**A**: *r*=0.33, 95% CI: –0.12 to 0.67, p=0.126), $rFE_{TQ}$ and VL fascicle length (**B**: *r*=0.63, 95% CI: 0.27–0.83, p=0.001), $rFE_{WS}$ and fascicle stretch amplitude (**C**: *r*=0.29, 95% CI: –0.19 to 0.66, p=0.201), and $rFE_{WS}$ and VL fascicle length (**D**: *r*=0.52, 95% CI: 0.08–0.79, p=0.017). Pearson correlations neglect the substantial within-subject variability and therefore were not performed (*Bakdash and Marusich, 2017*).

mean differences in absolute VL fascicle lengths were significant across the three muscle lengths for both contraction conditions (overall: 19 to 44 mm [95% CI: 8 to 56], p≤0.003). Significant repeated-measures linear relationships were found between absolute VL fascicle length (where mean fascicle lengths between the stretch-hold and fixed-end reference contractions were taken for each muscle length condition) and $rFE_{TQ}$ ($r$=0.63, 95% CI: 0.27 to 0.83, p=0.001), as well as absolute VL fascicle length and $rFE_{WS}$ ($r$=0.52, 95% CI: 0.08 to 0.79, p=0.017) (*Figure 2*).

### Kinematics

The mean target knee joint angles determined from the optical motion analysis system for the fixed-end reference contractions were 28.2±2.3°, 68.1±10.1°, and 83.6±7.9° for the short, long, and very long muscle lengths, respectively. For the stretch-hold contractions, the mean target knee joint angles were 28.2±4.3°, 68.7±18.4°, and 86.1±19.2° for the short, long, and very long muscle lengths, respectively. During the stretch-hold contractions, the measured knee joint rotations during the amplitude-matched dynamometer-imposed rotations were 9.0±4.3°, 14.7±2.3°, and 15.7±2.8° to the short, long, and very long muscle lengths, respectively. The respective programmed dynamometer rotations were 14.8±0.4°, 15.4±0.4°, and 15.2±0.3°.

## Discussion

The main purpose of this study was to examine whether rFE magnitudes, which were estimated from corrected knee extension torques and squared PT shear-wave speeds during time- and EMG-matched submaximal stretch-hold and fixed-end reference contractions, are different at short, long, and very long muscle lengths. To our knowledge, this is the first in vivo experiment that has attempted to estimate rFE at more than two muscle lengths, and the first to use shear-wave tensiometry. We found that corrected knee extension torques were significantly higher during the steady-state phase of the stretch-hold compared with the fixed-end reference contractions at both the long and very long muscle lengths, whereas squared PT shear-wave speeds were significantly higher only at the very long muscle length. For both $rFE_{TQ}$ and $rFE_{WS}$ measures, we found the greatest rFE at the very long muscle length. We also found significant and strong positive repeated-measures relationships between absolute VL muscle fascicle length and $rFE_{TQ}$ and $rFE_{WS}$, but no significant repeated-measures relationships between VL muscle fascicle stretch amplitude and $rFE_{TQ}$ or $rFE_{WS}$. Taken together, $rFE_{TQ}$ and $rFE_{WS}$ findings indicate that in vivo rFE is maximised during submaximal voluntary contractions of the human quadriceps at a very long muscle length, and that in vivo rFE is more strongly correlated with VL muscle fascicle length than fascicle stretch amplitude.

### Corrected knee extension torques

By using dynamometry and motion capture to calculate corrected knee extension torques during submaximal stretch-hold and fixed-end reference contractions, we found significant $rFE_{TQ}$ at the long and very long muscle lengths, but not at the short muscle length (*Figure 1A*). This is in accordance with a previous study with a similar setup, where we also found significant rFE only at a long (10.7±5.5%), but not at a short (1.3±3.1%) muscle length (*Bakenecker et al., 2020*). Similar findings have been reported by *Shim and Garner, 2012* at short (no rFE) and long (4.7%) quadriceps' muscle lengths. In contrast, *Power et al., 2013* also found significant rFE at short quadriceps' muscle lengths (4–13%), but the rFE at long muscle lengths was significantly higher (7–20%). Unlike the aforementioned studies, which either tested at a common knee flexion angle of 40° knee flexion or confirmed that the short muscle length was situated on the ascending limb of the quadriceps' torque-angle relationship, *Power et al., 2013* tested at 60° knee flexion for their short muscle length condition, which would result in a longer muscle length and therefore significant rFE based on our data (*Figures 1 and 3*).

In vitro findings suggest that rFE is not only muscle-length dependent, but also stretch-amplitude dependent, whereby rFE generally increases with increasing stretch amplitude (*Edman et al., 1982*; *Edman et al., 1978*). However, the only two in situ studies we are aware of that systematically investigated how stretch amplitude affects rFE showed that rFE does not always increase with increasing stretch amplitude (*Bullimore et al., 2007*; *Hisey et al., 2009*). For example, *Bullimore et al., 2007* showed that rFE increases with increasing stretch amplitude until a critical amplitude (9 mm of stretch)

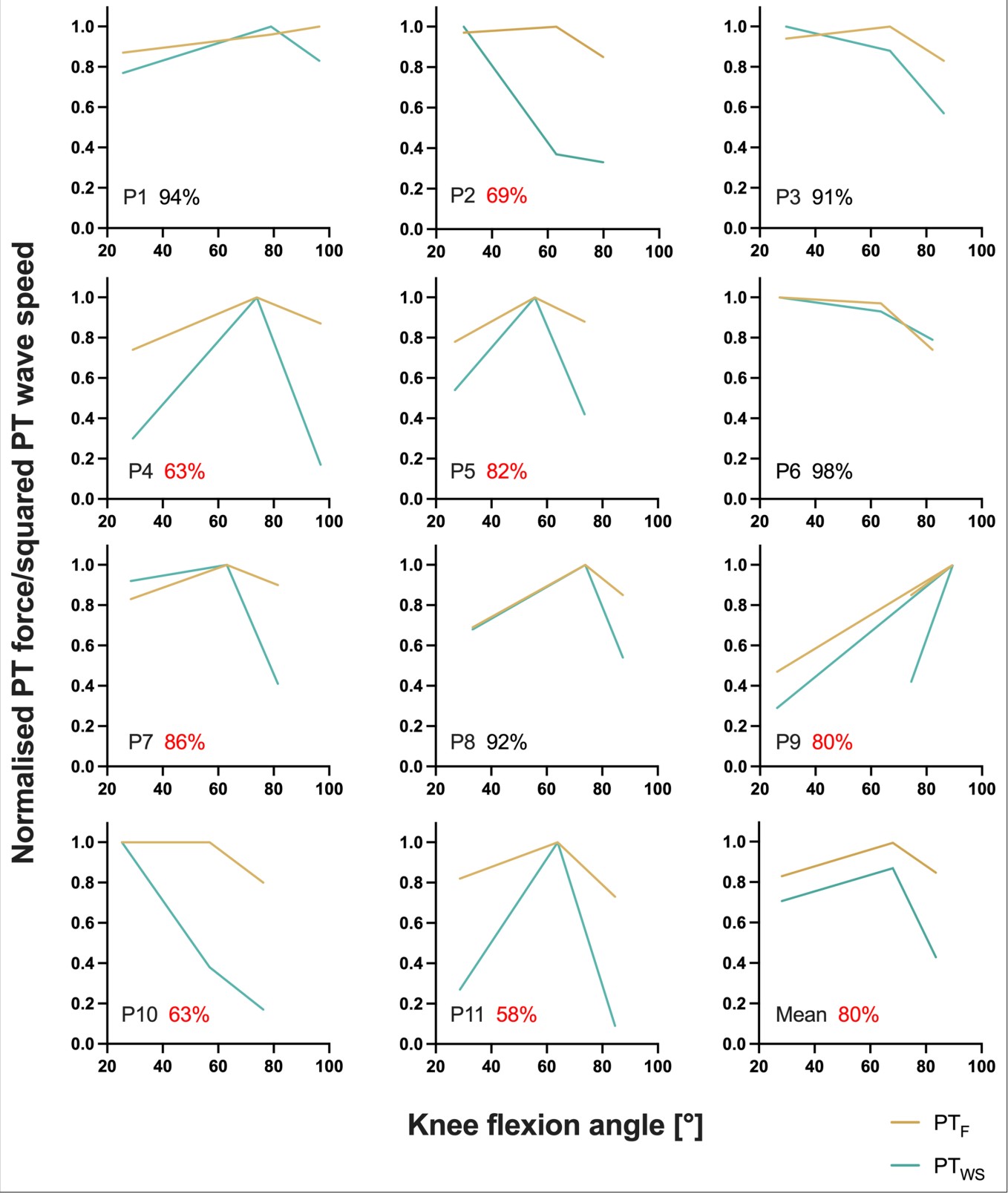

**Figure 3.** Individual normalised relationships between estimated patellar tendon (PT) force ($PT_F$) and knee flexion angle and between squared PT shear-wave speed ($PT_{WS}$) and knee flexion angle determined from the fixed-end reference contractions at the short, long, and very long muscle lengths from part 2 of the experiment. The average agreement across muscle lengths ($100 - (PT_{WS} - PT_F)/(PT_{WS} + PT_F) \times 100$) between normalised PT forces and squared PT shear-wave speeds is indicated as a percentage for every participant, where red font has been used to highlight instances of less than 90%

*Figure 3 continued on next page*

*Figure 3 continued*

agreement. The bottom right panel shows the mean normalised relationship between estimated PT force and knee flexion angle and between squared PT shear-wave speed and knee flexion angle across all participants. The angle-specific agreement across participants was 86%, 91%, and 62% at the short, long, and very long muscle lengths, respectively.

The online version of this article includes the following figure supplement(s) for figure 3:

**Figure supplement 1.** Individual normalised relationships between estimated patellar tendon (PT) force (PT$_F$) and knee flexion angle and between squared PT shear-wave speed (PT$_{WS}$) and knee flexion angle determined from the fixed-end reference ramp contractions at the short, long, and very long muscle lengths from part 1 of the experiment.

**Figure supplement 2.** Individual squared patellar tendon (PT) shear-wave-speed-time (green), knee extension torque-time (blue), and summed quadriceps' muscle activity level-time traces (grey) during maximum voluntary fixed-end reference contractions at a 70° crank-arm angle from part 1 of the experiment.

when the stretch ends on the descending limb of the force-length relationship. *Hisey et al., 2009* later showed that rFE only increases with increasing stretch amplitude until a critical amplitude (12 mm of stretch) when the stretch ends at very long muscle fascicle lengths on the descending limb of the force-length relationship (i.e. 9 mm longer than the optimum length). As we observed no significant repeated-measures linear relationship between VL fascicle stretch amplitude and rFE$_{TQ}$ (*Figure 2*), it is unlikely that greater stretch amplitudes contributed to greater rFE at the long and very long muscle lengths compared with the short muscle length. Based on the findings of *Hisey et al., 2009*, where increasing stretch amplitude did not increase the amount of rFE at a target muscle length 3 mm shorter than the muscle's optimal length for maximum isometric force production, it is further unlikely that greater stretch amplitudes to the short muscle length (i.e. on the ascending limb of the force-length relationship) would have changed the rFE values we observed. In contrast to in situ findings on the descending limb of the force-length relationship (*Hisey et al., 2009*), a recent meta-analysis by *de Campos et al., 2022* suggested that the rFE estimated from in vivo experiments is not affected by differences in stretch amplitude. Even though *de Campos et al., 2022* simply defined stretch amplitude based on the dynamometer-imposed rotation amplitude, this supports our finding that in vivo rFE does not seem to be stretch-amplitude dependent. In contrast, we found a significant and strong positive repeated-measures relationship between absolute VL muscle fascicle length and rFE$_{TQ}$ (*Figure 2*). This is in accordance with in situ experiments that showed rFE increases with increasing muscle length (*Bullimore et al., 2007*; *Hisey et al., 2009*). Therefore, our torque measures support the idea that in vivo rFE is more dependent on muscle length than stretch amplitude. The increase in rFE with muscle length could potentially be due to a greater increase in stiffness of a passive elastic structure, such as titin (*Rassier and Herzog, 2005*), as it is stretched to longer lengths.

## Shear-wave-speed measures

By using shear-wave tensiometry to assess PT loads during submaximal stretch-hold and fixed-end reference contractions, we found significant rFE$_{WS}$ only at the very long muscle lengths, but not at the short or long muscle lengths (*Figure 1B*). Therefore, our findings are only partly in accordance with other in vivo findings that tend to show greater rFE values at long compared with short muscle lengths within the human quadriceps (*Bakenecker et al., 2020*; *Power et al., 2013*; *Shim and Garner, 2012*) and plantar flexors (*Fukutani et al., 2017*; *Hahn et al., 2012*; *Hahn and Riedel, 2018*; *Pinniger and Cresswell, 2007*). With these previous findings in mind, we find it odd that rFE$_{WS}$ was prominent at the short muscle length (8%) even though the statistical analysis showed no significant difference in squared PT shear-wave speeds between stretch-hold and fixed-end contractions at this muscle length because of the rFE$_{WS}$ variability (12%). However, similar to rFE estimates based on corrected knee extension torques, we also found a significant and strong positive repeated-measures linear relationship between absolute VL muscle fascicle lengths and rFE$_{WS}$. This is in accordance with in situ findings from the cat soleus muscle, where rFE increased from <2% to ~2–4% to ~6–14% as the target muscle length increased from 3 mm shorter to 3 mm and 9 mm longer than the muscle's optimal length for isometric force production (*Hisey et al., 2009*). Therefore, our shear-wave-speed measures also support the idea that in vivo rFE is muscle-length dependent.

In this study, we found rFE magnitudes based on corrected knee extension torques of 2±5%, 7±5%, and 12±8% at the short, long, and very long muscle lengths, respectively. Based on PT shear-wave-speed measures, we observed rFE magnitudes of 8±12%, 6±10%, and 38±27% at the same respective

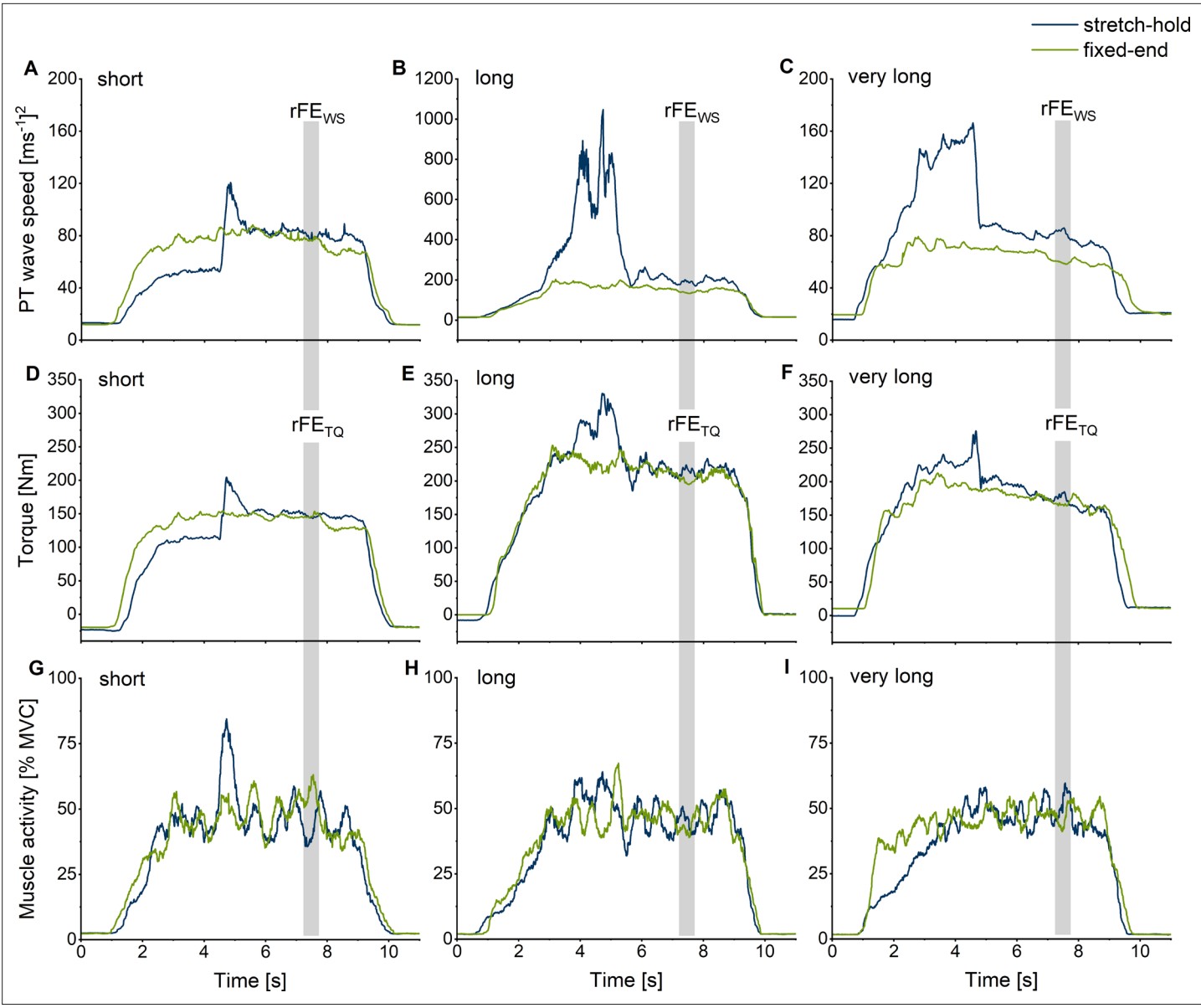

**Figure 4.** Exemplar (*n*=1) squared patellar tendon (PT) shear-wave-speed-time (**A, B, C**), corrected knee extension torque-time (**D, E, F**), and normalised quadriceps muscle activity level-time traces (**G, H, I**) for the stretch-hold (blue) and fixed-end reference (green) contractions at the short (**A, D, G**), long (**B, E, H**), and very long (**C, F, I**) muscle lengths. The vertical grey shaded areas show the time intervals where mean squared PT shear-wave speeds and corrected torques were analysed to evaluate $rFE_{WS}$ and $rFE_{TQ}$.

lengths. Except for the $rFE_{WS}$ values at the very long muscle length, the rFE values from this study are in accordance with findings from the study of *Seiberl et al., 2012*, which also had participants perform submaximal voluntary stretch-hold and fixed-end reference contractions of the human quadriceps. At a similar crank-arm angle (100° knee flexion) compared with what we used for the very long muscle length condition (104±6°), *Seiberl et al., 2012* observed rFE magnitudes of 7–11% and 3–5% at 30% and 60% of the maximum voluntary activity level, respectively. Other studies that investigated the human plantar flexors and the adductor pollicis showed similar rFE magnitudes (7–13%) during submaximal voluntary stretch-hold contractions (*Oskouei and Herzog, 2006*; *Oskouei and Herzog, 2005*; *Pinniger and Cresswell, 2007*). rFE magnitudes greater than ~15% have only been found for electrically stimulated human muscles (*Cook and McDonagh, 1995*). The $rFE_{WS}$ values above 25% (see *Figure 1B*) we calculated might therefore be erroneous, which is supported by the much lower than expected normalised squared PT shear-wave speeds determined during the fixed-end reference

contractions at the very long muscle length (see *Figures 3 and 4*). Unsurprisingly, the average agreement between normalised squared PT shear-wave speeds and estimated PT forces was worst at this muscle length at 62% compared with 86% and 91% at the short and long muscle lengths, respectively. This might be because the accuracy of the shear-wave-speed estimates, or PT force estimates, decreases as the PT's orientation changes relative to the tibia's long axis (*Draganich et al., 1987*), however this deserves further research.

In general, obtaining plausible shear-wave speeds from the PT proved to be very difficult. In some of our participants (*n*=5), the shear-wave tensiometry technique resulted in unusable data, which is not that surprising, considering that *Martin et al., 2018* also could not calculate shear-wave speeds from the PT during parts of the gait cycle (see their Supplementary Figure 6). For the remaining participants in the current study whose shear-wave speeds could be quantified, the accuracy of the measurements across different knee joint angles is questionable considering that we found an average agreement between squared PT shear-wave speeds and estimated PT forces above 90% for four participants only (*Figure 3* and *Figure 3—figure supplement 1*). PT shear-wave-speed measurements also became implausible during passive conditions where the MTU was at rest, during active conditions with stretch under high tension (*Figure 4B*), and during maximal voluntary contractions. Further, squared shear-wave speeds changed drastically across different joint angles, despite similar torque levels (*Figure 4B/E and C/F*). Taken together, this indicates that quantifying PT loads can be problematic with shear-wave tensiometry and that the PT shear-wave speeds estimated by this technique might be inaccurate.

The main assumption of shear-wave tensiometry is that the shear-wave propagation speed depends primarily on tendon axial stress under physiological loads (*Martin et al., 2018*). Even if this assumption holds true, which might not be the case under very low or high physiological loads (*Blank et al., 2022*; *Sarvazyan et al., 2013*), axial stress is influenced by both the force in the tendon and the tendon's architecture (i.e. the width and thickness of the tendon). A recent study by *Martin et al., 2020* suggests that the calibration gain required to estimate tendon stress based on tendon shear-wave speed appears to increase with increasing tendon aspect ratios (i.e. increasing tendon width relative to tendon thickness). Accordingly, when the tendon's architecture changes (e.g. through deformation of the tendon due to an applied stress) (*Kuervers et al., 2021*) under different contraction conditions (e.g. during a stretch-hold contraction that covers different joint angles and contraction intensities), shear-wave speeds could vary at the same tendon stress. Thus, shear-wave speeds may not accurately reflect tendon stress unless the changes in tendon architecture are accounted for.

The accuracy of the shear-wave-speed measures might also depend on the amount of stress that is applied to the tendon. In former studies (*Keuler et al., 2019*; *Martin et al., 2018*), high correlations between squared shear-wave speed ($\leq$80 $m^2$ $s^{-2}$) and torque measures ($\leq$200 Nm) were found when muscle activities and corresponding tendon stresses were low during preferred-speed walking ($\leq$50 MPa). In contrast, under the higher knee extension torques (112–296 Nm) and presumably higher tendon stresses (>50 MPa) at the submaximal muscle activity levels that were used in this study, the estimated PT force-angle relationships were not reflected by the squared shear-wave-speed measures (*Figure 3*). Additionally, squared shear-wave-speed estimates were unreasonably high when participants increased their activation level during the maximal voluntary contractions in the first part of the experiment (*Figure 3—figure supplement 2*). Therefore, high tendon stresses and changing tendon dimensions might be problematic for accurately estimating PT tendon loads based on shear-wave-speed measures.

Estimates of PT shear-wave speeds might additionally be affected by differences in subcutaneous tissue thickness between individuals as *Martin et al., 2018* showed that the propagating shear wave within the deep tendon was not identical to the shear wave propagating along the skin. Compared with the Achilles tendon, the PT has more overlying subcutaneous tissue (*Bravo-Sánchez et al., 2019*) and is smaller in thickness in its mid-region (*Coombes et al., 2018*). These factors might result in poorer shear-wave propagation within the PT compared with the Achilles tendon and less accurate shear-wave-speed estimates. However, further research is needed to investigate whether thicker subcutaneous tissue over a thinner tendon reduces the accuracy of tendon shear-wave-speed measurements made over the skin.

## Non-responders

Previous in vivo experiments that examined rFE during voluntary contractions have identified a number of participants not showing rFE (*Hahn et al., 2012*; *Hahn et al., 2007*; *Oskouei and Herzog, 2006*; *Tilp et al., 2009*). These participants were subsequently called non-responders. We also found instances where participants did not show enhanced torques and/or enhanced PT shear-wave speeds during the stretch-hold compared with the fixed-end reference contractions. However, in opposition to the idea of there being a true non-responder, there was no participant in our dataset who consistently showed no rFE across the three muscle lengths we tested. We thus speculate there is no true non-responder. This speculation is supported by recent work by *Paternoster et al., 2021*, who showed that voluntary rFE appeared inconsistently in the same participants across five testing sessions, whereas during submaximal tetanic electrical muscle stimulation, all participants consistently showed rFE. Indeed, having participants match muscle activity levels through voluntary effort, rather than through artificial stimulation, likely adds variability to rFE estimates (especially over a short duration analysis window like the one used here), and might contribute to the 'non-responder' phenomenon.

## Limitations

Although we performed feedback-controlled contractions based on EMG recordings to ensure the same level of superficial quadriceps' activity during stretch-hold and fixed-end reference contractions and we assessed VL muscle fascicle stretch magnitudes, our rFE values could be influenced by different preloads prior to the stretch phase of the stretch-hold contractions at the short, long, and very long muscle lengths. Due to the relatively high compliance of human lower limb MTUs, using a preload prior to a stretch-hold contraction at the MTU level results in shortening-isometric-stretch-isometric muscle fascicle behavior, where the initial shortening of muscle fascicles increases with an increasing preload or a shorter initial muscle length. As rFE can be influenced by the amount of shortening preceding stretch (*Lee et al., 2001*) and active shortening can induce residual force depression (i.e. a reduction in the steady-state force following shortening *Abbott and Aubert, 1952*; *Granzier et al., 1989*), differences in muscle fascicle shortening across the three tested muscle lengths could have influenced the estimated rFE magnitudes. Further, we only assessed stretch amplitudes and muscle fascicle lengths from a limited region within the VL muscle and we needed to perform linear extrapolation to estimate fascicle lengths, so we recommend that future studies investigate a muscle with shorter and less curved fascicles to better evaluate the influence of stretch amplitude and fascicle length on rFE.

## Conclusion

The purpose of this study was to investigate whether in vivo rFE within the human quadriceps varies with muscle length. We therefore investigated rFE within the human quadriceps and PT at short, long, and very long muscle lengths by respectively using conventional dynamometry and motion analysis (rFE$_{TQ}$) and a new, non-invasive technique known as shear-wave tensiometry (rFE$_{WS}$). We found significant rFE$_{TQ}$ at the long (7±5%) and very long (12±8%) muscle lengths, but not at the short (2±5%) muscle length, whereas rFE$_{WS}$ was only significant at the very long (38±27%) muscle length, but not at the short (8±12%) or long (6±10%) muscle lengths. We also found significant and strong positive repeated-measures relationships between absolute VL muscle fascicle length and rFE$_{TQ}$ and rFE$_{WS}$, but no significant relationships were observed between VL muscle fascicle stretch amplitude and rFE$_{TQ}$ or rFE$_{WS}$. We conclude that increasing muscle length, rather than increasing stretch amplitude, contributes more to in vivo rFE during submaximal voluntary contractions of the human quadriceps. Additionally, the low agreement (<90%) across tested muscle lengths between normalised squared PT shear-wave speeds and normalised PT forces estimated from corrected knee extension torques suggests that assessing PT loads with shear-wave tensiometry might be inaccurate.

# Materials and methods

## Participants

Sixteen healthy males (age 27.9±5.3 years; height 186.3±5.6 cm; weight 83.4±8.6 kg) gave free written informed consent prior to participating in the study, 11 of whom (age 28.4±5.9 years; height 187.1±5.8 cm; weight 82.8±8.0 kg) had data that could be analysed as their PT shear-wave speeds

could be calculated. The recruited participants were relatively tall to increase the chances of successfully securing the tendon tapper device over a longer PT. As pilot testing revealed that individuals should be at least 181 cm, we tested males only because we were not able to recruit any female participants that were this tall. All participants were free of knee injuries and neuromuscular disorders. All experimental procedures were approved by the local Ethics Committee of the Faculty of Sport Science at Ruhr University Bochum and conformed with the Declaration of Helsinki.

## Experimental protocol

The entire experiment consisted of one familiarisation session and one test session that each consisted of two parts. Familiarisation helped participants to become comfortable with the dynamometer and improved their ability to perform maximum voluntary contractions (MVCs) and EMG-matched stretch-hold and fixed-end contractions. Each experimental session started with a standardised warm-up that included 10–15 submaximal knee extensions (~50–80% of perceived maximum effort) to precondition the MTU (*Maganaris et al., 2002*). MVCs were performed with standardised verbal encouragement from the investigator, while participants received real-time visual feedback of their net knee joint torque (*Gandevia, 2001*). During EMG-matched contractions, participants received real-time (250 ms delay) visual feedback of their 500 ms moving root-mean-square (RMS) summed quadriceps' muscle activity amplitude (i.e. muscle activities of VL, RF, and VM were summed), which they attempted to visually match within predefined traces that were 5% apart.

## Test session part 1 – Determination of the relationship between shear-wave speed and knee flexion angle

Participants' individual relationships between shear-wave speed and knee flexion angle were first determined to ensure that the stretch-hold and corresponding fixed-end reference contractions during the second part of the experiment were performed at short (i.e. ascending limb of the quadriceps' force-angle relationship), long, and very long (i.e. both on the descending limb of the quadriceps' force-angle relationship) muscle lengths. To determine the relationships between shear-wave speed and knee flexion angle, participants performed maximum voluntary knee extension contractions at 30°, 70°, and 110° crank-arm angles (*Figure 5*). Subsequently, participants performed EMG-matched ramp contractions from rest to 50% of their angle-specific maximum muscle activity level at the same crank-arm angles. Participants were instructed to build up to the required 50% muscle activity level over a period of 3 s and then to hold this level for another 2 s. To minimise fatigue, participants received at least 3 min and 2 min of rest between MVCs and submaximal contractions, respectively.

## Test session part 2 – Determination of rFE at short, long, and very long muscle lengths

Using the submaximal (50% of maximal activity) relationship between shear-wave speed and knee flexion angle determined in part 1 of the experiment, one knee joint angle on both sides of the plateau of the relationship between shear-wave speed and knee flexion angle with the same PT shear-wave speed (i.e. 85% of the maximum recorded shear-wave speed) was selected as the reference knee joint angle for the fixed-end contractions at short and long muscle lengths. An additional fixed-end reference contraction was performed at a knee joint angle that was 15° more flexed than the joint angle at the long muscle length (which is referred to as the 'very long muscle length'). Stretch-hold contractions to the short, long, and very long muscle lengths were performed with a 15° crank-arm rotation amplitude to the predetermined knee joint angle for each muscle length condition. The hold phase of the stretch-hold contractions was 6 s, which allowed rFE to be determined (*Figures 4 and 6*). The crank-arm rotation velocity was set to 60°s⁻¹ and was triggered 1.5 s after participants reached the plateau of the required muscle activity level. Similar to part 1, participants were asked to ramp up their muscle activity level and hold it at 50% of their angle-specific maximum muscle activity level during stretch-hold and fixed-end contractions (*Figure 6A*). The two contraction types were performed at least three times each in a randomised order and participants received at least 2 min of rest between contractions. At the end of the session, participants performed a maximum knee flexion contraction at a 70° crank-arm angle, and the maximum EMG activity amplitude was used to normalise the biceps femoris muscle activity level during the stretch-hold and fixed-end reference knee extension contractions.

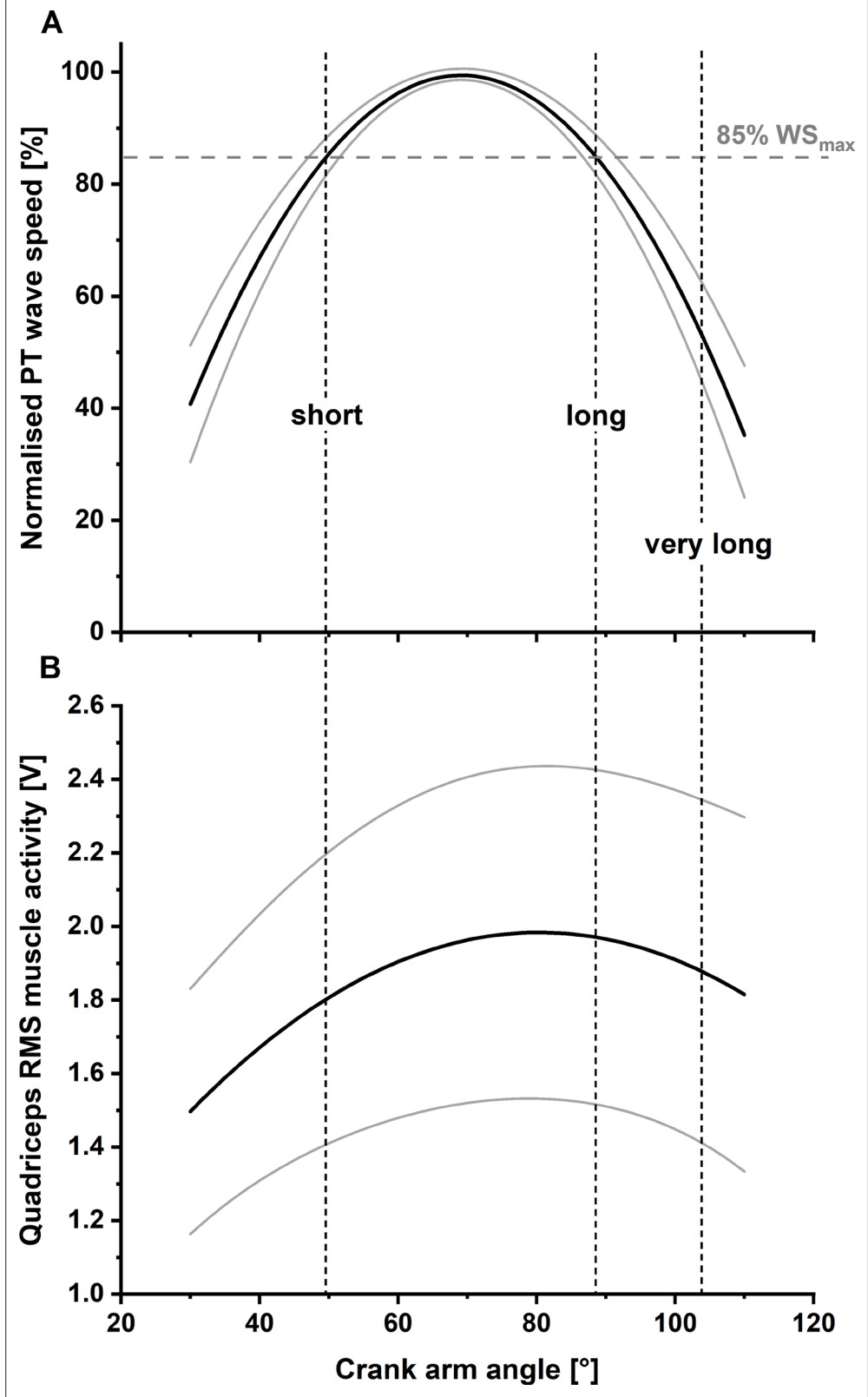

**Figure 5.** Solid lines indicate the mean normalised fitted relationship between patellar tendon (PT) shear-wave speed and knee flexion angle (Panel **A**) and mean fitted relationship between quadriceps' muscle activity level and knee flexion angle (Panel **B**) across all participants (*N*=11) with lower and upper 95% confidence intervals (dashed lines). Two different knee joint angles (i.e. the short and long muscle lengths), with a matched PT shear-wave-speed

*Figure 5 continued on next page*

*Figure 5 continued*

capacity (85% of maximum PT shear-wave speed), were selected as the target knee joint angles for the stretch–hold and fixed-end reference contractions. A third target knee joint angle, referred to as the 'very long muscle length', was defined as a crank-arm angle 15° more flexed than the crank-arm angle at the long muscle length.

## Experimental setup

Participants performed knee extension contractions with their right leg while sitting in a reclined position (100° hip flexion) on the seat of a motorised dynamometer (IsoMed2000, D&R Ferstl GmbH, Hemau, Germany). Each participant's right lower leg was fixed with Velcro around the mid-shank to a cushioned attachment that was connected to the crank arm of the dynamometer and their foot was vertically aligned with their knee. Two shoulder restraints and one waist strap were used to secure participants firmly in the seat of the dynamometer. Participants folded their arms across their chest prior to each contraction to limit accessory movements. To compensate for the knee joint rotation that occurs during activation of the quadriceps (*Arampatzis et al., 2004*; *Bakenecker et al., 2019*), the knee joint axis and dynamometer axis of rotation were aligned during an active contraction of the quadriceps at every final knee joint position tested (i.e. corresponding to the short, long, and very long muscle lengths). The knee and dynamometer axes were aligned with the help of a laser pointer that projected the dynamometer axis onto the skin, and the position and orientation of the dynamometer axis were adjusted until the laser was located over the palpated lateral femoral condyle and in line with the presumed axis of rotation of the knee.

## Torque measurements

The dynamometer was used to measure net knee joint torque and crank-arm angle during stretch-hold and fixed-end contractions. Torque and crank-arm angle data was sampled at 2 kHz and synchronised using a 16-bit Power 1401 and Spike2 data collection system (Cambridge Electronic Design, UK).

## PT shear-wave speed

Similar to *Martin et al., 2018*, we determined PT shear-wave speed using a piezo-actuated tapper that delivered micron-scale impulses through the skin over the PT, which induced a transient shear wave (i.e. a transverse motion of the tendon) that was detected by two adjacent miniature accelerometers. The mechanical tapping device was built as described in *Martin et al., 2018*. The tapper had a lever arm of 10 mm and its axis of rotation was secured over the PT using adhesive bandage. The 16-bit Power 1401 produced a square-wave pulse sequence (0.5–9.5 V, 50% duty cycle, 50 Hz repetition rate) that was amplified by an open-loop piezo controller (MDT694B, Thorlabs, Newton, USA) used to drive the tapper. Transient shear waves were induced in the PT by both extension and retraction of the actuator, but shear-wave speeds were derived only from the extension of the actuator, which resulted in 50 Hz shear-wave-speed data.

The transverse motion of the PT was measured at two points using single-axis accelerometers (PCB-352A26, PCB Synotech, Germany) with 10 mV/g sensitivity. The miniature (8.6 × 4.1 × 2.8 mm³, 0.2 g) accelerometers were secured in tight-fitting silicon housings and strapped over the PT using adhesive tape. The first accelerometer was positioned approximately 5 mm proximal to the tapper. The inter-accelerometer distance (centre-to-centre) was 9 mm and accelerometer signals were sampled at 100 kHz using the Spike2 data collection system. PT shear-wave speed was calculated following each contraction using Python and the steps previously described in *Martin et al., 2018*.

## Knee joint kinematics

Despite our best efforts to ensure the knee joint and dynamometer crank arm rotated in the same plane, translation of the knee joint axis relative to the dynamometer axis of rotation can occur due to knee joint movement during contraction. Therefore, we determined the actual knee joint angles and corrected knee extension torque measurements for axes misalignment using optical motion capture data (OptiTrack, NaturalPoint, Oregon, USA) collected with four cameras. Three reflective markers were positioned on the cushioned attachment that was connected to the crank arm of the dynamometer (i.e. the mid-shank), with one marker each on the most prominent points of the lateral and medial femoral epicondyles, and on the line between the greater trochanter and the lateral femoral

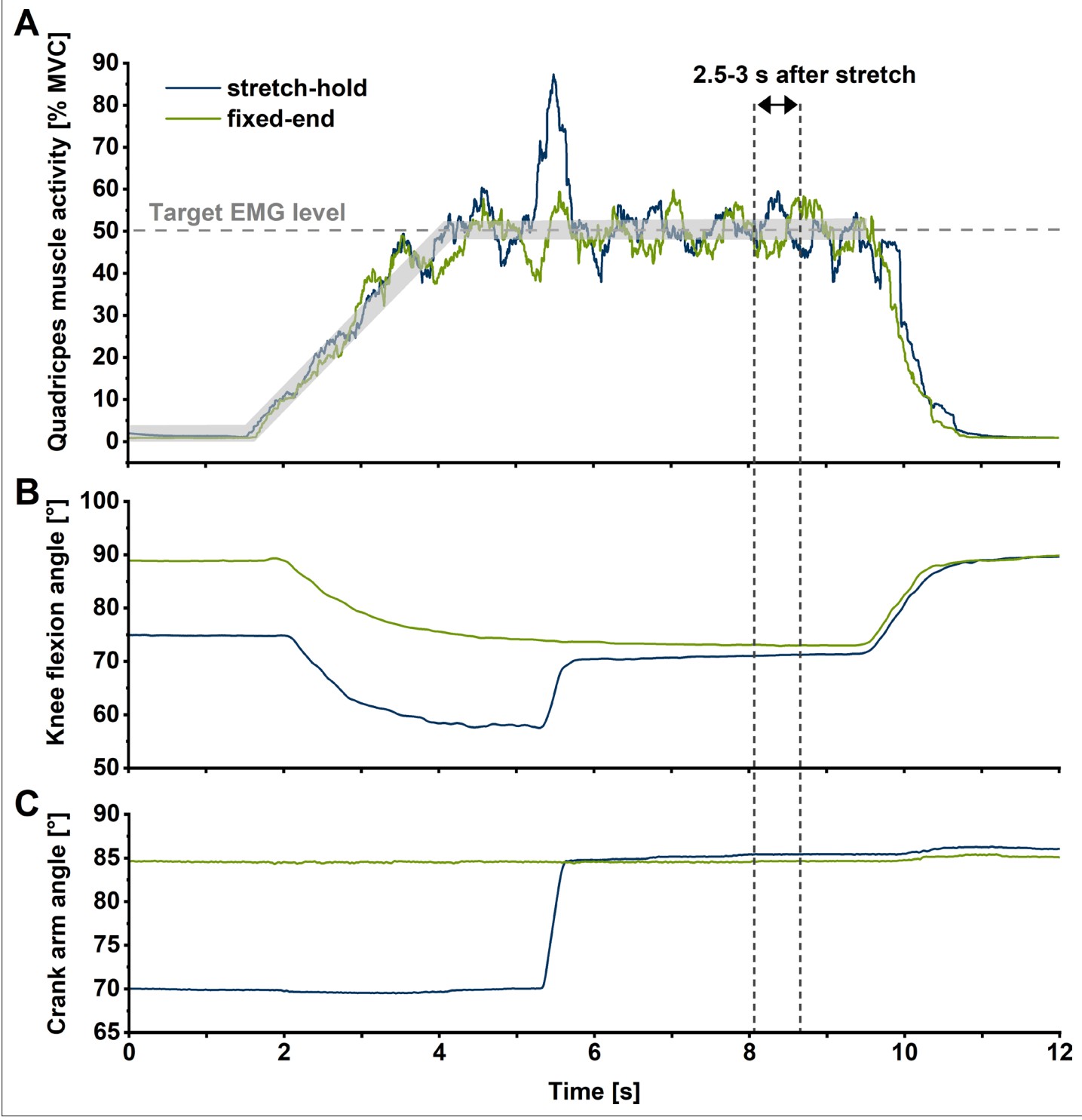

**Figure 6.** Exemplar (*n*=1) summed superficial quadriceps' muscle activity level-time (**A**), knee flexion angle-time (**B**), and crank-arm angle-time traces during stretch-hold (blue line) and fixed-end reference (green line) contractions at the long muscle length. For each muscle length condition, participants were instructed to match their quadriceps' muscle activity level (calculated as a 250 ms centred root-mean-square amplitude) between two predefined traces 5% apart that ramped up to 50% of their angle-specific maximum over 3 s during both stretch-hold and fixed-end reference contractions. Outcome measures were analysed in the time interval from 2.5 to 3 s after stretch (vertical dotted lines), which corresponded to ~6 s after contraction onset in the stretch-hold and fixed-end reference contractions.

condyle. To determine potential misalignment of the knee joint axis and the dynamometer axis, three additional markers were placed on the dynamometer, with one marker over the dynamometer axis (*Figure 7*). Data was sampled at 100 Hz using Motive software (version 2.2, OptiTrack, NaturalPoint, Oregon, USA) and synchronised with all other data by a common digital pulse.

### Surface electromyography

Surface EMG (NeuroLog System NL905, Digitimer Ltd, UK) was used to record the muscle activities of the VL, RF, VM, and biceps femoris of the right leg. After skin preparation (i.e. shaving, abrading, and swabbing the skin with antiseptic), two surface electrodes (8 mm recording diameter, Ag/AgCl, H124SG, Kendall, Mansfield, Massachusetts, USA) were placed over these muscles according to SENIAM guidelines (*Hermens et al., 2000*) using a 2 cm inter-electrode distance. Due to the placement of the ultrasound transducer, electrodes were placed towards the distal end of VL's mid-belly. A single reference electrode was secured over the fibular head of the left leg. EMG signals were band-pass filtered between 0.01 and 20 kHz and amplified 2000 times (NL844, Digitimer Ltd, UK), before being sampled at 2 kHz using the Spike2 data collection system.

### Ultrasound imaging

To image the muscle fascicles of VL during stretch-hold and fixed-end contractions, a PC-based ultrasound system (LogicScan 128 CEXT-1Z Kit, Telemed, Vilnius, Lithuania) was used, which was connected to a flat-sided 96-element transducer (LV7.5/60/96, B-mode, 8.0 MHz, 60 mm depth; Telemed, Vilnius, Lithuania). The transducer captured images at 150 Hz and was placed over the mid-belly of VL (*Sharifnezhad et al., 2014*). The location of the transducer on the skin was secured using a custom 3D-printed plastic frame and adhesive bandage. The ultrasound system generated a digital pulse that was used to synchronise all digital signals to a common start and end time.

### Data analysis

All data processing and analysis were performed using custom-written scripts in Python (scripts and exemplary data can be found at the following repository link: https://github.com/NeuromecHAHNics/Bakenecker_et_al_2022; copy archived at swh:1:rev:72c97dde7c5a53dd1f908caa4317fd29b-d268562;*Bakenecker, 2022*). Torque and crank-arm angle data was filtered using a dual-pass fourth-order 20 Hz low-pass Butterworth filter. The knee angle was determined as the dihedral angle between the planes of the three markers on the shank and thigh. EMG signals from the superficial quadriceps muscles (VL, RF, VM) and the biceps femoris muscle were smoothed using a centred moving RMS amplitude calculation of 250 ms. Calculated shear-wave speed was squared. Fascicle length data was calculated using filtered x-y fascicle endpoint coordinates, which were filtered with a dual-pass second-order 6 Hz low-pass Butterworth filter. Only trials that had less than 6% difference between the predefined angle-specific EMG target level and measured superficial quadriceps' EMG level were included in the analysis (*Raiteri and Hahn, 2019*).

### Corrected torque

To account for potential over- or underestimation of the knee extension torque due to misalignment between the knee joint axis and the dynamometer axis, we recalculated knee extension torque as follows. The point of force application by the shank was assumed to be at the centre of the cushioned pad attached to the shank. From this centre point, the lever arm of the dynamometer crank arm was measured as the shortest perpendicular distance to the axis of rotation of the dynamometer. The lever arm of the shank was measured as the perpendicular distance between the knee joint centre (assumed to be halfway between the medial and lateral femoral epicondyles) and the centre of the cushioned shank pad. Furthermore, the dihedral angle between the crank arm and the shank was determined (*Figure 7*). The corrected torque was calculated by first finding the force at the centre of the cushioned shank pad through division of the measured peak-to-peak torque by the dynamometer lever arm. This force was then divided by the cosine of the angle between the crank arm and shank to find the force that was applied perpendicular to the shank. Corrected torque was then this corrected force multiplied by the lever arm of the shank.

### Relationship between shear-wave speed and knee flexion angle

For each tested knee joint angle, PT shear-wave speed was normalised to the maximum shear-wave speed obtained during the submaximal fixed-end reference contractions at the three target knee joint

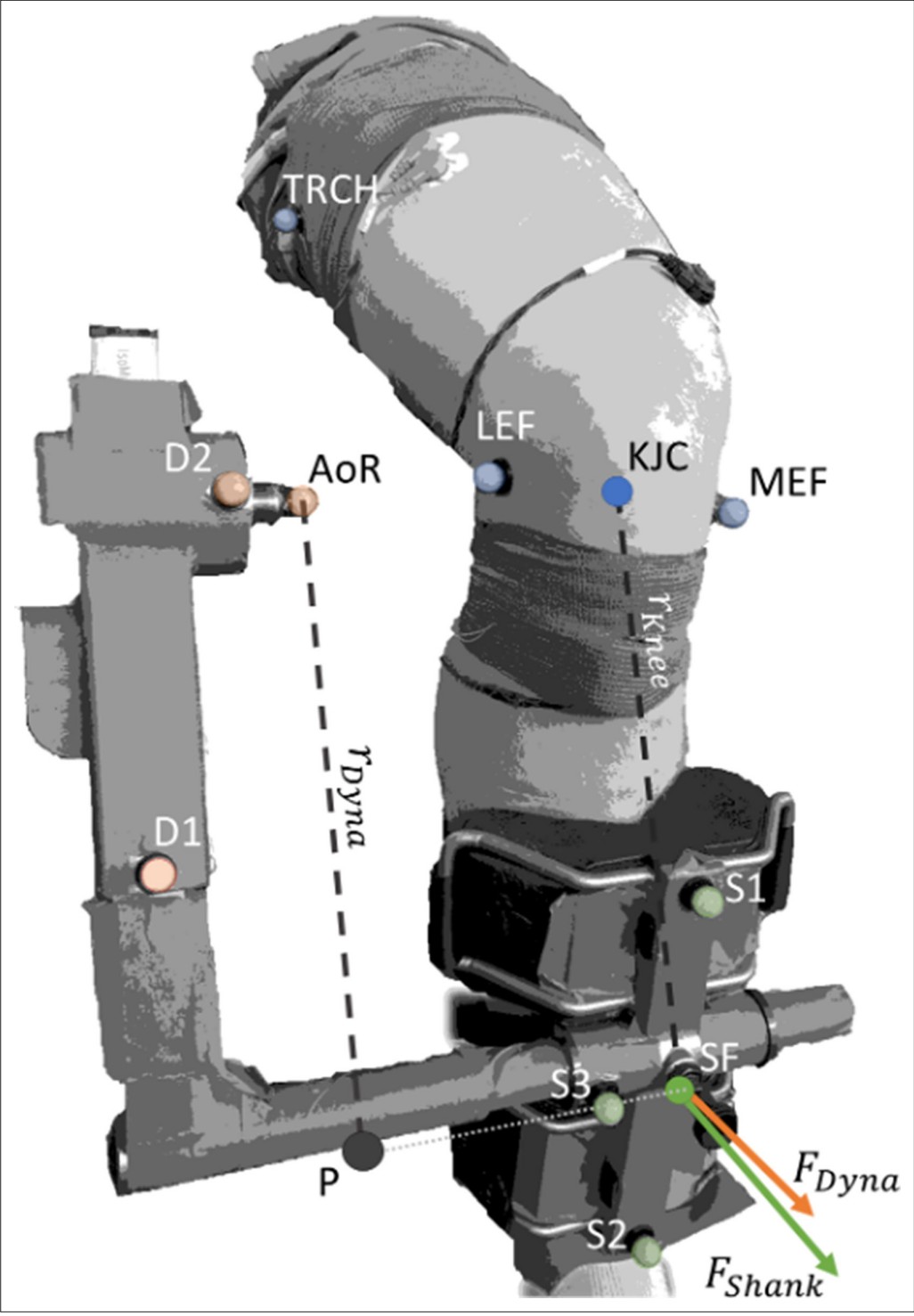

**Figure 7.** Corrected torque was calculated by multiplying the force acting at the shank ($F_{Shank}$) by the external moment arm to the knee ($r_{Knee}$). $F_{Shank}$ acted at the midpoint of the pad attached to the shank and $r_{Knee}$ was the perpendicular distance from the line of action of $F_{Shank}$ to the knee joint centre (KJC). $F_{Shank}$ was calculated by dividing the calculated dynamometer force ($F_{Dyna}$) by the cosine of the angle between the two force vectors, $F_{Shank}$ and $F_{Dyna}$, whose respective directions were defined by the normal vectors of the planes formed by Shank Marker 1 (**S1**), S2, and S3, and Dynamometer Marker 1 (**D1**), D2, and the dynamometer's axis of rotation (AoR). $F_{Dyna}$ was calculated by dividing the measured torque at the dynamometer AoR by the external moment arm of the dynamometer ($r_{Dyna}$). $r_{Dyna}$ was calculated as the distance between P and AoR. P was defined as the shortest

*Figure 7 continued on next page*

*Figure 7 continued*

distance between the projection of AoR onto the vector formed by S3 and the midpoint of the pad attached to the shank (SF). Transparent colored markers indicate captured markers, whereas solid marker KJC was calculated as the midpoint between the lateral and medial epicondyles of the femur (LEF and MEF, respectively), and solid marker SF was calculated as the midpoint between S1 and S2.

angles. Following this, a second-order polynomial curve was fitted to the shear-wave-speed crank-arm angle data.

rFE based on corrected torque ($rFE_{TQ}$) and shear-wave-speed ($rFE_{WS}$) measurements was calculated as the percent mean difference between the stretch-hold and fixed-end reference contractions from 2.5 to 3.0 s after stretch (i.e. ~6 s after contraction onset) at the same target knee joint angle. The 0.5 s interval to calculate rFE used here is in line with previous studies (*Hahn et al., 2012*; *Hahn et al., 2010*) and ensures that rFE was estimated during the steady-state following stretch.

## Muscle activity level

Superficial quadriceps' muscle activities and biceps femoris muscle activity during stretch-hold contractions were quantified by taking the mean EMG RMS amplitude from 2.5 to 3 s after stretch. Muscle activity during stretch-hold contractions was then compared with the time-matched muscle activity during the fixed-end contractions. Mean EMG RMS values of VL, RF, and VM, and the summed superficial quadriceps' activity level during stretch-hold and fixed-end contractions were then normalised to the angle-specific maximum muscle activity level recorded during the MVCs. To determine the desired EMG levels that participants needed to match at the target knee joint angles (i.e. short, long, and very long muscle lengths), we fitted a second-order polynomial curve to the EMG-angle data from the MVCs at 30°, 70°, and 110° crank-arm angles.

## Fascicle length and length changes

VL fascicle lengths from each muscle length condition were initially calculated offline from the last image of each ultrasound recording using linear extrapolation to find the intersections between one representative fascicle and VL's superficial and deep aponeuroses. To reduce subjectivity of the fascicle length determination, the fascicle orientation was automatically determined by fitting a straight-line through user-selected feature points, and the aponeuroses were automatically determined using feature detection. Custom software was then used to track VL muscle fascicle length changes from the last to first ultrasound image by applying a Lucas-Kanade-Tomasi-based affine optic flow algorithm. This algorithm is similar to the algorithm used by UltraTrack (*Farris and Lichtwark, 2016*), except that it only tracks detected feature points (*Shi and Tomasi, 1994*) across sequential images. To improve algorithm performance, the feature points were only detected from within a user-defined region of interest that spanned the image field of view and was located on VL's superficial aponeurosis and just above (i.e. 0.4–3.5 mm) VL's deep aponeurosis. Additionally, ultrasound recordings were downsampled from 150 to 30 fps as this resulted in less frame-to-frame fascicle length change underestimation.

To determine whether fascicle lengths were similar between the stretch-hold and fixed-end reference contractions, VL muscle fascicle length data was analysed during the same time interval as the torque and shear-wave-speed data. During stretch-hold contractions, the magnitude of fascicle stretch was determined from the start to end of fascicle lengthening 0.5 s before and after the dynamometer-imposed rotation. The start of fascicle lengthening was determined as the first time point when VL fascicle velocity was ≥0.05 mms$^{-1}$, and the end of fascicle lengthening was determined as the last time point when VL fascicle velocity was ≥0.05 mms$^{-1}$. VL fascicle stretch was then calculated by calculating the difference in fascicle length at these two time points. VL fascicle velocity was determined by differentiating the filtered fascicle length data.

## PT force

To estimate in vivo PT force, the corrected knee extension torques during the fixed-end contractions at short, long, and very long muscle lengths were divided by the angle-specific PT moment arms from the recommended mean PT moment arm function provided by *Bakenecker et al., 2019*. The agreement between the squared PT shear-wave speeds and estimated PT forces at each muscle length during the fixed-end contractions was then calculated by subtracting the symmetrised percent

difference (*Nuzzo, 2018*) between normalised squared PT shear-wave speeds and normalised PT forces from 100. For example, if peak squared PT shear-wave speed (PT$_{WS}$) and peak estimated PT force (PT$_F$) occurred at the same knee flexion angle, then both normalised values would be equal to 1 and the agreement would be:

$$100 - (PTWS - PTF) \div (PTWS + PTF) \times 100 = 100 - (1 - 1) \div (1+1) \times 100 = 100\%$$

The symmetrised percent difference was used because subtracting a percent difference from 100 resulted in negative (i.e. <0%) agreement when the mean normalised value was less than the absolute difference. Squared PT shear-wave speeds and PT forces were normalised to their maximum during the fixed-end reference submaximal contractions over the three target joint angles.

## Statistics

To detect outliers in muscle activity, squared shear-wave speed, torque, or fascicle length data in the stretch-hold or fixed-end reference conditions, a method based on a range of four around the MAD was used (*Leys et al., 2013*). Two-way repeated-measures ANOVAs or mixed-effects analyses (only for datasets with missing values due to outliers) were performed to identify differences in summed superficial quadriceps' muscle activity levels, corrected knee extension torques, squared PT shear-wave speeds, and absolute VL fascicle lengths between stretch-hold and fixed-end reference contractions across muscle lengths (contraction condition × muscle length). A two-way repeated-measures ANOVA was also used to identify differences in rFE between rFE methods (rFE$_{TQ}$ and rFE$_{WS}$) across muscle lengths (rFE determination method × muscle length). A one-way repeated-measures ANOVA was used to identify differences in VL fascicle stretch magnitudes across muscle lengths during the stretch-hold contractions. Repeated-measures Pearson correlation coefficients were calculated to test the strength of the relationships between rFE$_{TQ}$ or rFE$_{WS}$ and VL fascicle stretch amplitudes or absolute VL fascicle lengths using the rmcorr package (*Bakdash and Marusich, 2017*) in RStudio (v1.4.1717, 2021, Boston, Massachusetts, USA). The alpha level was set at 5% and statistical analysis was performed using commercially-available software (GraphPad Prism 9, San Diego, California, USA).

## Additional information

### Funding
No external funding was received for this work.

### Author contributions
Patrick Bakenecker, Conceptualization, Writing – original draft, acquisition of data; data analysis; data interpretation; Tobias Weingarten, Conceptualization, Writing – review and editing, acquisition of data; data analysis; data interpretation; Daniel Hahn, Conceptualization, Writing – review and editing, data interpretation; Brent Raiteri, Conceptualization, Writing – review and editing, data analysis ; data interpretation

### Author ORCIDs
Patrick Bakenecker http://orcid.org/0000-0002-3938-2806
Daniel Hahn http://orcid.org/0000-0002-9401-5478
Brent Raiteri http://orcid.org/0000-0002-2078-9075

### Ethics
Participants gave free written informed consent prior to participating in the study. All experimental procedures were approved by the local Ethics Committee of the Faculty of Sport Science at Ruhr University Bochum and conformed with the Declaration of Helsinki.

### Decision letter and Author response
Decision letter https://doi.org/10.7554/eLife.77553.sa1
Author response https://doi.org/10.7554/eLife.77553.sa2

## Additional files

### Supplementary files
• Transparent reporting form

### Data availability
The final processed data can be found at: https://figshare.com/s/d66e2c7400480dd0a059. It provides source data of figures 1, 2, 3, 4, 5 and 6.

The following dataset was generated:

| Author(s) | Year | Dataset title | Dataset URL | Database and Identifier |
|---|---|---|---|---|
| Bakenecker P, Weingarten T, Hahn D, Raiteri BJ | 2022 | Data_Residual force enhancement within the human quadriceps is greatest during submaximal stretch-hold contractions at a very long muscle length | https://figshare.com/s/ d66e2c7400480dd0a059 | figshare, d66e2c7400480dd0a059 |

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
