## [Editor Report]

This manuscript will be of interest to those who study human performance and muscle physiology. The study involved a very careful evaluation of a phenomenon whereby eccentric contraction increases the force generating capacity of skeletal muscle. Several modern techniques including measurements of muscle fascicle length and kinematics were used. The results sharpen our understanding of the relationships between muscle length and performance.

---

## [Decision Letter]

**Decision letter after peer review:**

Thank you for submitting your article "Residual force enhancement within the human quadriceps is greatest during submaximal stretch-hold contractions at a very long muscle length" for consideration by *eLife*. Your article has been reviewed by 2 peer reviewers, including Christopher Cardozo and Reviewing Editor and Reviewer #1, and the evaluation has been overseen by Mone Zaidi as the Senior Editor. The following individual involved in review of your submission has agreed to reveal their identity: Jonathan Jarvis (Reviewer #2).

Essential revisions:

1) Please cite a reference for the Python code used if it is already publicly available; otherwise please deposit the code in a repository and provide a link or include the code in supplementary material.

2) Please address the comments of Reviewer 2 regarding Figures 1 and 5.

3) Please carefully consider the remaining comments from both reviewers.

*Reviewer #1 (Recommendations for the authors):*

1. Demographics of research participants such as age, height, weight should be added. Influence of gender as a biological variable should be discussed. If there is no data on effect of gender on force potentiation this might be discussed as an open question for the field.

2. Any anthropomorphic data pertinent to the study such as those that affect calculation of the lever arm for each participant would strengthen the study. Examples might include individual variations in bone lengths or distance between femoral condyles.

3. Python code should be uploaded to a repository or a link to a repository if not already publicly available. If the code is publicly available a link or literature citation should be included.

4. Page 28 under Experimental setup: is the idea here to be sure that all joint forces are in the same plane as that in which the lever arm on the dynamometer rotates? Please clarify.

5. My understanding is that distance between the tapper and transducers is a variable in estimating PT shear wave speed. Was this variable controlled for? Please elaborate in methods as appropriate and discuss any limitations of the technique's application to human subjects. When doing so, please comment at least in the response to critique as to whether skin thickness or mechanical properties are variables that must be considered when interpreting data using this method.

6. Under Joint Kinematics (starting line 566) it would help the reader if the cartoon shown in supplemental figures was referenced in this section.

7. To help the reader who is not well read in physiology please explain what stretch-hold and fixed-end contractions are in a way that the non-expert could understand.

8. It would be helpful to the field to describe some of the challenges in employing shear wave velocity for this purpose. Consider discussing how variability in tendon architecture could be captured such as by MRI imaging as a means to better control these measurements in the future. Consider discussing the relationship between load and conduction speed which I gather from the paper by Martin is related to the square root of load. If that is true, then sensitivity to changes in load diminishes with increasing load.

9. Consider discussing a bit more why you were or were not surprised that stretch did not change residual force and compare/contrast to any ex-vivo or in-vitro experiments so that the reader and draw conclusions about whether the lack of relationship in your study relates to inherent variability and reduced sensitivity of physiological measurements in humans.

10. Please explain in more detail what is shown in Figure 2. It looks like the relationship is plotted for each subject. If that is the case, consider showing all data points but showing only the lines for correlation and 95% confidence intervals on this graph.

11. I commend the authors on the care taken to align the knee relative to the dynamometer. It was not clear to me how the leg was positioned to assure the alignment of the tibia relative to the knee and crank arm of the dynamometer was performed to assure that knee extension was in plane with the movement of the crank arm. Additional explanation of how this would be done would probably help the reader.

*Reviewer #2 (Recommendations for the authors):*

Figure 5 shows the force estimated with the shear wave technique, and the muscle emg signal, as they vary with muscle length (knee angle) in isometric contractions. The figure would helpfully also include the force estimated from the measured torque. Otherwise the reader cannot tell whether the curve shown is the expected shape representing the ascending and descending limbs of the force-length curve, or a non-linearity of the shear wave measurement. This presentation would also have allowed a comparison of the two methods of force estimation without the influence of movement.

There are also two versions of Figure 1 with apparently the same data but different values for the variability estimates. Maybe I missed an explanation here. But in any case, if there is any non-linearity between actual muscle force and the shear wave estimate, then the variability calculated as the difference between the normalised estimates divided by the mean of the estimates may not be very meaningful. What is noticeable in Figure 1 is that the peak is always at the intermediate knee angle for both measures.

line 38 unique has no meaning here.

line 40 does fixed-end mean isometric in your paper? If not it would be useful to explain why not since you also use isometric in the text.

line 62 please explain operating range – do you mean 'over the normal range of motion?

line 70. Could you cite evidence that the peak of the force length curve occurs at significantly different knee angles for different individuals?

line 114 need to explain why 90%, and what you were comparing. was it the estimate of force based on shear wave speed or was it the shear wave speed? Also, you need to say rather than 'the force-length relationship', the measured force based on dynamometer torque across the length range.

In general the use of noun-phrases with hyphens is confusing. For example, in the phrase 'shear-wave-speed-angle relationship' you need to indicate that the hyphen between speed and angle means something different than the rest. It is clearer to say 'the relationship between shear wave speed and the knee angle'.

272 'increases more' is ambiguous. Better to say 'is better correlated with fascicle length than fractional fascicle stretch'.

line 283 (7-20%) should be at the end of the sentence otherwise it looks like it refers to muscle length rather than rFE.

line 351. If these are average agreements, then are the numbers used in Figure 3 average agreements?

line 394 I think you need to explain what unreasonable means here. As in my earlier comments, you need to take a view on whether the shear wave measurement is good in isometric but not dynamic contractions, or whether it is just difficult to manage. same in line 449 (where do these numbers come from? I can't see them in results.).

---

## [Author Response]

Essential revisions:1) Please cite a reference for the Python code used if it is already publicly available; otherwise please deposit the code in a repository and provide a link or include the code in supplementary material.

The Python code has now been made available at the following repository link: https://github.com/NeuromecHAHNics/Bakenecker_et_al_2022

We have indicated the above information within the manuscript here (lns 644-646):

“All data processing and analysis were performed using custom-written scripts in Python (scripts and exemplary data can be found at the following repository link: https://github.com/NeuromecHAHNics/Bakenecker_et_al_2022).”

2) Please address the comments of Reviewer 2 regarding Figures 1 and 5.

Please see our responses below to Reviewer 2’s comments.

3) Please carefully consider the remaining comments from both reviewers.

We thank the editor and the reviewers for their constructive feedback and for the time and effort they spent reviewing our manuscript. We have addressed all comments on a point-by-point basis below.

While revising our manuscript, we decided to change the title from “Residual force enhancement within the human quadriceps is greatest during submaximal stretch-hold contractions at a very long muscle length” to “Residual force enhancement is affected more by quadriceps muscle length than stretch amplitude”. We think that the new title is more concise and still reflects our main finding and hope that the editor and reviewers agree.

Reviewer #1 (Recommendations for the authors):1. Demographics of research participants such as age, height, weight should be added. Influence of gender as a biological variable should be discussed. If there is no data on effect of gender on force potentiation this might be discussed as an open question for the field.

The reviewer must have missed the participant demographics information as it is within the methods and materials section. Nevertheless, we did not provide the demographics of all tested participants, so this has now been added to the manuscript (lns 481-484):

“Sixteen healthy males (age 27.9 ± 5.3 years; height 186.3 ± 5.6 cm; weight 83.4 ± 8.6 kg) gave free written informed consent prior to participating in the study, eleven of whom (age 28.4 ± 5.9 years; height 187.1 ± 5.8 cm; weight 82.8 ± 8.0 kg) had data that could be analysed as their PT tendon shear-wave speeds could be quantified.”

It was not our initial intention to test males only, however pilot testing revealed that it was difficult to secure the tendon tapper over the patellar tendon of shorter (<1.81 m) individuals. We thus aimed to recruit taller participants and it was easier to find taller males than females. This is why we ended up testing males only and why our participant group had an average height of 1.87 m. We have added this information to the manuscript (lns 484-487):

“The recruited participants were relatively tall to increase the chances of successfully securing the tendon tapper device over a longer patellar tendon. As pilot testing revealed that individuals should be at least 181 cm, we tested males only because we were not able to recruit any female participants that were this tall.”

The representative datasets from Martin et al., (2018) that show patellar tendon and biceps femoris tendon shear-wave speeds during treadmill running are also from male participants. Rather than males having better quality tendon shear-wave-speed data than females, we think that the chances of successfully obtaining subjectively good patellar tendon shear-wave-speed data increases as participant height exceeds 181 cm, and as males are generally taller than females, it is easier to collect better quality data from males. Because gender affects anthropometrics, and anthropometrics likely affect the quality of patellar tendon shear-wave-speed data, we would prefer not to speculate about how gender alone could affect our results. Additionally, other literature on in vivo residual force enhancement comes from both males and females and no gender effect has been demonstrated.

2. Any anthropomorphic data pertinent to the study such as those that affect calculation of the lever arm for each participant would strengthen the study. Examples might include individual variations in bone lengths or distance between femoral condyles.

We did not assess individual participant characteristics like bone length or the distance between the femoral condyles because of previous research that shows scaling patellar tendon moment arm to anthropometric characteristics is inaccurate (Tsaopoulos et al., 2007). Further, between-subject moment arm differences would not affect our repeated-measures rFE results. Within-subject moment arm differences between stretch-hold and fixed-end contraction conditions are likely minor as dynamometer crank arm angles were matched between conditions and knee joint torques were submaximal and similar (maximum mean difference: <22 Nm, see Table 1).

3. Python code should be uploaded to a repository or a link to a repository if not already publicly available. If the code is publicly available a link or literature citation should be included.

The Python code has now been made available at the following repository link:

https://github.com/NeuromecHAHNics/Bakenecker_et_al_2022

We have indicated the above information within the manuscript here (lns 644-646).:

“All data processing and analysis were performed using custom-written scripts in Python (scripts and exemplary data can be found at the following repository link: https://github.com/NeuromecHAHNics/Bakenecker_et_al_2022).”

4. Page 28 under Experimental setup: is the idea here to be sure that all joint forces are in the same plane as that in which the lever arm on the dynamometer rotates? Please clarify.

Yes, our aim was to ensure that the plane the knee joint rotated in was identical to the plane that the dynamometer crank arm rotated in. However, as it is possible to have rotation in the same plane, but around a translated axis, we decided to align the axes of rotation of the knee joint and dynamometer as closely as possible when the torque measurement was analysed. We then used our collected motion capture data to correct the measured net knee joint torques for axes misalignment. We have made the following changes within the manuscript to make this clear (lns 570-577 and lns 607-611):

“To compensate for the knee joint rotation that occurs during activation of the quadriceps (Arampatzis et al., 2004; Bakenecker et al., 2019) and to ensure that the knee joint and dynamometer crank arm rotated in the same plane, the knee joint axis and dynamometer axis of rotation were aligned during an active contraction of the quadriceps at every final knee joint position tested (i.e. corresponding to the short, long, and very long muscle lengths). The knee and dynamometer axes were aligned with the help of a laser pointer that projected the dynamometer axis onto the skin, and the position and orientation of the dynamometer axis were adjusted until the laser was located over the palpated lateral femoral condyle and in line with the presumed axis of rotation of the knee.”

Knee joint kinematics

Despite our best efforts to ensure the knee joint and dynamometer crank arm rotated in the same plane, translation of the knee joint axis relative to the dynamometer axis of rotation can occur due to knee joint movement during contraction. Therefore, we determined the actual knee joint angles and corrected net joint torque measurements for axes misalignment using optical motion capture data (Optitrack, Natural Point, OR, USA) collected with four cameras.

5. My understanding is that distance between the tapper and transducers is a variable in estimating PT shear wave speed. Was this variable controlled for? Please elaborate in methods as appropriate and discuss any limitations of the technique's application to human subjects. When doing so, please comment at least in the response to critique as to whether skin thickness or mechanical properties are variables that must be considered when interpreting data using this method.

If the tapping device is relatively far away from the accelerometers and the transient shear waves are not detected by the relatively more distal accelerometer, then shear-wave speeds cannot be estimated. To our knowledge, this is the only case where the distance between the tapper and accelerometers would affect the estimation of patellar tendon shear-wave speeds. However, transient shear waves were detected by both accelerometers in our study even though we did not standardize the distance between the tapper and accelerometers. The distance between the two accelerometers does affect the shear-wave speed estimations, but the accelerometers were secured in a housing with a fixed and known distance of 9 mm.

We previously discussed how tendon architecture and the mechanical conditions might affect tendon shear-wave speed estimates in lines 393-405, but the reviewer raises an important point that skin thickness (as well as subcutaneous tissue thickness) might also affect shear-wave speed estimates. Based on Figure 2 of Martin et al., (2018), transverse Achilles tendon velocity decreases not only with an increasing distance from the tapper, but also with a decreasing depth to the skin. Consequently, a greater tissue thickness between the tendon and the accelerometers could cause greater attenuation of the propagating shear waves, which could decrease the accuracy of the shear-wave speed measurements. Differences in subcutaneous tissue thickness between individuals might therefore have affected the accuracy of our tendon shear-wave speed estimates. To our knowledge, the difficulties we faced with estimating patellar tendon shear-wave speeds have not been discussed in previous studies that estimated Achilles tendon shear-wave speeds with the same technique. Therefore, we did not expect to experience difficulties and these may have arisen because the patellar tendon has more overlying subcutaneous tissue (Bravo-Sánchez et al., 2019) and is thinner in its mid-portion than the Achilles tendon (Coombes et al., 2018). Both factors might result in poorer shear-wave propagation within the patellar tendon and thus greater shear-wave speed measurement error. We think this deserves further research and so we have made the following addition to the manuscript (lns 419-427):

“Estimates of PT shear-wave speeds might additionally be affected by differences in subcutaneous tissue thickness between individuals as Martin et al., (2018) showed that the propagating shear wave within the deep tendon was not identical to the shear wave propagating along the skin. Compared with the Achilles tendon, the PT has more overlying subcutaneous tissue (Bravo-Sánchez et al., 2019) and is thinner in its mid-region (Coombes et al., 2018). These factors might result in poorer shear-wave propagation within the PT compared with the Achilles tendon and less accurate shear-wave speed estimates. However, further research is needed to investigate whether thicker subcutaneous tissue over a thinner tendon reduces the accuracy of tendon shear-wave speed measurements made over the skin.”

Tendon shear-wave speed estimates could also be affected by the mechanical properties of the tendon. The patellar tendon is shorter and larger in cross-section than the Achilles tendon (Mogi, 2020), which results in a relatively stiffer patellar tendon. Consequently, shear waves should propagate faster within the patellar tendon at the same tendon force, and this would require a faster accelerometer sampling rate to accurately estimate the shear-wave speed. With our sampling rate, we estimate that we could detect patellar tendon shear-wave speeds of up to 450 m/s (i.e. squared shear-wave speeds of 202 500 m^2^s^-2^), which is much higher than the values we report in the manuscript.

Other tendon mechanical properties, such as stress relaxation or creep might have affected our tendon shear-wave speed estimates, however because our contraction conditions were time-matched, shear-wave speed estimates from the steady-state of submaximal contractions should have been influenced similarly.

6. Under Joint Kinematics (starting line 566) it would help the reader if the cartoon shown in supplemental figures was referenced in this section.

Thanks for bringing this to our attention. We have now added this reference (lns 615-617). Please note that Figure 3 of the supplementary figures is now Figure 7:

“…To determine potential misalignment of the knee joint axis and the dynamometer axis, three additional markers were placed on the dynamometer, with one marker over the dynamometer axis (Figure 7) ….”

7. To help the reader who is not well read in physiology please explain what stretch-hold and fixed-end contractions are in a way that the non-expert could understand.

Thanks for bringing up this potential point of confusion. We have made the following addition to the introduction (lns 108-111):

“Stretch-hold contractions indicate that the quadriceps muscle-tendon unit was actively stretched by a triggered rotation of the dynamometer crank arm before being held at a constant length. Fixed-end contractions indicate that the quadriceps muscle-tendon unit length remained relatively constant for the duration of the contraction.”

8. It would be helpful to the field to describe some of the challenges in employing shear wave velocity for this purpose. Consider discussing how variability in tendon architecture could be captured such as by MRI imaging as a means to better control these measurements in the future. Consider discussing the relationship between load and conduction speed which I gather from the paper by Martin is related to the square root of load. If that is true, then sensitivity to changes in load diminishes with increasing load.

Based on an earlier comment, we now discuss that factors such as subcutaneous tissue thickness could have affected the accuracy of the tendon shear-wave speed estimates. Assessing tendon architecture would be useful if one wants to estimate tendon stress from a tendon shear-wave speed measurement, which we have already indicated in lns 400-405. However, our results show that tendon shear-wave speeds estimated from shear-wave tensiometry can be inaccurate. We simply do not believe that these inaccurate measurements could be made more accurate by normalising to tendon dimensions.

Based on the equations put forward in the Martin et al., (2018) paper, the sensitivity to changes in load should increase with increasing load, as tendon shear-wave speed squared is linearly related to tendon axial stress. We found the largest tendon shear-wave speed differences occurred between stretch-hold and fixed-end reference conditions at the very long muscle length, which is not when the assumed tendon axial stress would have been the highest (see updated Table 1). If the difference in tendon axial stress would have been equivalent between the stretch-hold and fixed-end reference conditions across the three tested muscle lengths, then the difference in squared shear-wave speed should have been largest at the long muscle length because tendon axial stress was presumably highest at this length.

9. Consider discussing a bit more why you were or were not surprised that stretch did not change residual force and compare/contrast to any ex-vivo or in-vitro experiments so that the reader and draw conclusions about whether the lack of relationship in your study relates to inherent variability and reduced sensitivity of physiological measurements in humans.

It is typically cited that rFE increases with increasing stretch amplitude (e.g., Groeber et al., 2020), but the two in vitro studies that systematically investigated how stretch amplitude affects rFE show that this is generally not the case (Bullimore et al., 2007; Hisey et al., 2009). rFE only increases with increasing stretch amplitude at longer muscle lengths than the optimum length for maximum isometric force production (Hisey et al., 2009). At shorter lengths than optimum, increasing stretch amplitude can actually abolish rFE (Hisey et al., 2009). Therefore, our in vivo data agree with the in vitro data despite the inherently greater measurement error and thus reduced sensitivity of in vivo measurements. We have made the following addition within the manuscript to make this clear (lns 298-308):

*“*in vitro findings suggest that rFE is not only muscle-length dependent, but also stretch-amplitude dependent, where rFE generally increases with increasing stretch amplitude (Edman et al., 1982, 1978). However, the only two in vitro studies we are aware of that systematically investigated how stretch amplitude affects rFE showed that rFE does not always increase with increasing stretch amplitude (Bullimore et al., 2007; Hisey et al., 2009). For example, Bullimore et al., (2007) showed that rFE increases with increasing stretch amplitude until a critical amplitude (9 mm of stretch) when the stretch ends on the descending limb of the force-length relationship. Hisey et al., (2009) later showed that rFE only increases with increasing stretch amplitude until a critical amplitude (12 mm of stretch) when the stretch ends at very long muscle fascicle lengths on the descending limb of the force-length relationship (9 mm longer than the optimum length).”

Note that we already mentioned within the manuscript that our in vivo data largely agrees with the in vitro data discussed above (lns 320-324):

“…we found a significant and strong positive repeated-measures relationship between absolute VL muscle fascicle length and rFE_TQ_ (Figure 2). This is in accordance with in vitro experiments that showed rFE increases with increasing muscle length (Bullimore et al., 2007; Hisey et al., 2009).”

10. Please explain in more detail what is shown in Figure 2. It looks like the relationship is plotted for each subject. If that is the case, consider showing all data points but showing only the lines for correlation and 95% confidence intervals on this graph.

The reviewer is correct that Figure 2 shows linear fits for each subject. The fits represent repeated-measures linear relationships between rFE_TQ_ / rFE_WS_ and vastus lateralis muscle fascicle length / stretch amplitude. We prefer repeated-measures (i.e. within-subject) correlations to Pearson (i.e. between-subject) correlations because the latter method requires the data from the same participant to be averaged to avoid violating the assumption of independence. Bakdash and Marusich (2017) showed that averaging participant data can produce misleading results if there are meaningful individual differences. There were meaningful individual differences in rFE_TQ_ / rFE_WS_, as well as fascicle length and stretch amplitude (see Figure 2), and neglecting this within-subject variability decreases our statistical power and the strength of the linear correlations. Therefore, we would prefer to keep the figure as it is. We have added the following to the figure legend to help justify our statistical analysis (lns 239-240):

“Pearson correlations neglect the substantial within-subject variability and therefore were not performed (Bakdash and Marusich, 2017).”

11. I commend the authors on the care taken to align the knee relative to the dynamometer. It was not clear to me how the leg was positioned to assure the alignment of the tibia relative to the knee and crank arm of the dynamometer was performed to assure that knee extension was in plane with the movement of the crank arm. Additional explanation of how this would be done would probably help the reader.

We have already made additions to the manuscript to explain how we ensured the knee joint and dynamometer crank arm rotated in the same plane in response to your fourth comment. When we fixed the participant’s shank to the cushioned pad projecting from the dynamometer crank arm, we ensured that their foot and knee were vertically aligned. We have added this information to the manuscript (lns 565-567):

“The participants’ right lower leg was fixed with Velcro around the mid-shank to a cushioned attachment that was connected to the crank arm of the dynamometer and their foot was vertically aligned with their knee.”

Reviewer #2 (Recommendations for the authors):Figure 5 shows the force estimated with the shear wave technique, and the muscle emg signal, as they vary with muscle length (knee angle) in isometric contractions. The figure would helpfully also include the force estimated from the measured torque. Otherwise the reader cannot tell whether the curve shown is the expected shape representing the ascending and descending limbs of the force-length curve, or a non-linearity of the shear wave measurement. This presentation would also have allowed a comparison of the two methods of force estimation without the influence of movement.

Figure 5 shows the normalized patellar tendon shear-wave speed on the y-axis, not the estimated patellar tendon force. This is a methodology figure and should help the reader to understand how the target knee joint angles and target EMG levels were selected before testing was performed at the three different muscle lengths. We would therefore like to avoid including estimated force-angle relationships in this figure because we did not use this relationship to select the target knee joint angles and target EMG levels. The reviewer’s mentioned comparison of the two methods of force estimation is already shown for each participant in figure 1 of the supplementary material. We agree with the reviewer that the comparison of tendon shear-wave speed and estimated force is important for the reader to see, which is why we have included this information in Figure 3—figure supplement 1 (bottom right panel). Descriptive statistics of the shear-wave speed estimates and patellar tendon force estimates at the three muscle lengths for both contraction conditions are also provided in Table 1.

There are also two versions of Figure 1 with apparently the same data but different values for the variability estimates. Maybe I missed an explanation here. But in any case, if there is any non-linearity between actual muscle force and the shear wave estimate, then the variability calculated as the difference between the normalised estimates divided by the mean of the estimates may not be very meaningful. What is noticeable in Figure 1 is that the peak is always at the intermediate knee angle for both measures.

We assume that the reviewer is referring to figure 3 within the manuscript and figure 1 within the supplementary material (now Figure 3—figure supplement 1), which show different data and thus have different values for the variability estimates. Figure 3 shows the normalized patellar tendon force estimates and normalized patellar tendon shear-wave speed estimates during the fixed-end reference contractions at the tested short, long, and very long muscle lengths (second part of the experiment). Figure 1 within the supplementary material (now Figure 3—figure supplement 1) shows the normalized patellar tendon force estimates and normalized patellar tendon shear-wave speed estimates during fixed-end contractions at common knee flexion angles of 30°,70° and 110° from the first part of the experiment, which allowed us to estimate individual force-angle relationships. This information is already provided in the figure legends.

Although the peak normalized patellar tendon force and peak normalized shear-wave speed was not *always* at the intermediate knee flexion angle for both measures, the reviewer is right that providing an average agreement obscures the differences in agreement across the tested muscle lengths. We have now also provided the average agreement across participants at each tested muscle length (Figure 3) and the average agreement across participants at each common knee flexion angle that we tested (Figure 3—figure supplement 1). Please refer to our response to your last comment to see the changes to the manuscript text.

line 38 unique has no meaning here.

Thanks for pointing this out. We have removed the term unique.

line 40 does fixed-end mean isometric in your paper? If not it would be useful to explain why not since you also use isometric in the text.

Thanks for bringing up this potential point of confusion. All instances of isometric refer to when muscle force was constant (i.e. at a steady state) and the quadriceps’ muscle-tendon unit length and muscle fascicle lengths were therefore relatively constant. All instances of fixed-end refer to when the dynamometer crank arm was not triggered to rotate, which resulted in a relatively constant quadriceps’ muscle-tendon unit length, but muscle fascicle shortening as force increased before the target EMG level was reached. As also recommended by reviewer 1, we have now explained the term fixed-end in the introduction (lns 110-111):

“Fixed-end contractions indicate that the quadriceps muscle-tendon unit length remained relatively constant for the duration of the contraction.”

We have also added “isometric” in line 40 to make this clear:

“Residual force enhancement (rFE) is a history-dependent property of skeletal muscle and is defined as the enhanced isometric (i.e. steady-state) force following active muscle lengthening (i.e. stretch) relative to the isometric force obtained during a fixed-end reference contraction at the same muscle length and level of activation (Cook and McDonagh, 1995; Edman et al., 1978).”

line 62 please explain operating range – do you mean ‘over the normal range of motion?

We mean the muscle’s operating region in relation to its optimum length for maximum isometric force production. We have made the following change within the manuscript (lns 61-65):

“However, this review neglected the previously reported in vitro based interaction between stretch amplitude and muscle length on rFE (Hisey et al., 2009), largely because very few in vivo studies have estimated the operating region of an individual muscle of interest in relation to its optimum length for maximum isometric force production.”

line 70. Could you cite evidence that the peak of the force length curve occurs at significantly different knee angles for different individuals?

Yes, please refer to Table 1 from Bakenecker et al., (2019). We have added this reference to the manuscript text (lns 70-74):

“One potential reason for the conflicting results could be that most studies tested at identical joint angles across participants for the different muscle length conditions, even though this could have resulted in rFE being examined only over the ascending *or* descending limb of the force-angle relationship in some participants, and over both ascending *and* descending limbs in other participants (Bakenecker et al., 2019).”

line 114 need to explain why 90%, and what you were comparing. was it the estimate of force based on shear wave speed or was it the shear wave speed? Also, you need to say rather than 'the force-length relationship', the measured force based on dynamometer torque across the length range.

The definition of a strong agreement (≥90%) was based on the results of Martin et al., (2018). We have now included this reference in the text. We have also changed the wording in line with the reviewer’s suggestion (lns 119-122):

“Based on the findings of Martin et al., (2018), we also expected a strong agreement (≥90%) between squared PT shear-wave speed and the PT force estimated from the resultant knee joint torque.”

In general the use of noun-phrases with hyphens is confusing. For example, in the phrase 'shear-wave-speed-angle relationship' you need to indicate that the hyphen between speed and angle means something different than the rest. It is clearer to say 'the relationship between shear wave speed and the knee angle'.

We agree and have changed the wording, for example (lns 506-514):

Test session part 1 – Determination of the relationship between shear-wave speed and knee flexion angle

“Participants’ individual relationships between shear-wave speed and knee flexion angle were first determined to ensure that the stretch-hold and corresponding fixed-end reference contractions during the second part of the experiment were performed at short (i.e. ascending limb of the F-l-r), long, and very long (i.e. both on the descending limb of the F-l-r) muscle lengths. To determine the relationships between shear-wave speed and knee flexion angle, participants performed maximal voluntary knee extension contractions at 30°, 70°, and 110° crank arm angles (Figure 5).”

For further corrections please see the revised manuscript with tracked changes.

272 'increases more' is ambiguous. Better to say 'is better correlated with fascicle length than fractional fascicle stretch'.

We agree and have changed the wording to (lns 278-281):

“Taken together, rFE_TQ_ and rFE_WS_ findings indicate that in vivo rFE is maximised during submaximal voluntary contractions of the human quadriceps at a very long muscle length, and that in vivo rFE is more strongly correlated with vastus lateralis muscle fascicle length than fascicle stretch amplitude.”

line 283 (7-20%) should be at the end of the sentence otherwise it looks like it refers to muscle length rather than rFE.

We agree and have changed the wording (lns 290-292):

“In contrast, Power et al., (2013) also found significant rFE at short quadriceps muscle lengths (4–13%), but the rFE at long muscle lengths was significantly higher (7–20%).”

line 351. If these are average agreements, then are the numbers used in Figure 3 average agreements?

These numbers refer to average agreements for each muscle length across participants, whereas the numbers in figure 3 show the average agreement for each participant across muscle lengths. To make this clear, we changed the wording to (lns 364-367):

“Unsurprisingly, the average agreement between normalised squared PT shear-wave speeds and estimated PT forces was worst at this muscle length at 62% compared with 86% and 91% at the short and long muscle lengths, respectively.”

line 394 I think you need to explain what unreasonable means here. As in my earlier comments, you need to take a view on whether the shear wave measurement is good in isometric but not dynamic contractions, or whether it is just difficult to manage. same in line 449 (where do these numbers come from? I can't see them in results.).

Unreasonable here means that the estimated squared shear-wave speeds are unreasonably high based on the assumed tendon stress. We have changed the wording to (lns 413-416):

“Additionally, shear-wave speed estimates were unreasonably high when participants increased their activation level during the maximal voluntary contractions in the first part of the experiment (Figure 3—figure supplement 2).”

In ln 449, the numbers were rFE_WS_ and rFE_TQ_ standard deviations, which were provided to indicate the greater variability of the rFE_WS_ estimates compared with the rFE_TQ_ estimates. However, as this was not clear, we have removed these numbers. Based on the agreement between normalised PT shear-wave speeds and normalised PT forces at each tested muscle length across participants, PT shear-wave speed estimates were more accurate at the long muscle length (91% agreement) compared with the short (86% agreement) and very long (62% agreement) muscle lengths. This could be because the orientation of the patellar tendon affects the shear-wave speed estimates (and/or the patellar tendon force estimates) and these estimates might be most accurate when the patellar tendon’s orientation is parallel to the quadriceps’ tendon’s orientation. Although speculative, we have now added this possibility to the text (lns 364-369) and we have made the following conclusion (lns 475-478):

“Unsurprisingly, the average agreement between normalised squared PT shear-wave speeds and estimated PT forces was worst at this muscle length at 62% compared with 86% and 91% at the short and long muscle lengths, respectively. This might be because the accuracy of the shear-wave speed estimates, or PT force estimates, decreases as the PT’s orientation changes relative to the tibia’s long axis (Draganich et al., 1987), however this deserves further research.”

“Additionally, the low agreement (<90%) across tested muscle lengths between normalised PT shear-wave speeds and normalised PT forces estimated from resultant knee joint torques suggests that assessing PT loads with shear-wave tensiometry might be inaccurate.”

References

Arampatzis A, Karamanidis K, de Monte G, Stafilidis S, Morey-Klapsing G, Brüggemann G-P. 2004. Differences between measured and resultant joint moments during voluntary and artificially elicited isometric knee extension contractions. *Clinical Biomechanics* 19:277–283. doi:10.1016/j.clinbiomech.2003.11.011

Bakdash JZ, Marusich LR. 2017. Repeated Measures Correlation. *Front Psychol* 8:456. doi:10.3389/fpsyg.2017.00456

Bakenecker P, Raiteri B, Hahn D. 2019. Patella tendon moment arm function considerations for human vastus lateralis force estimates. *Journal of Biomechanics* 86:225–231. doi:10.1016/j.jbiomech.2019.01.042

Bravo-Sánchez A, Abián P, Jiménez F, Abián-Vicén J. 2019. Myotendinous asymmetries derived from the prolonged practice of badminton in professional players. *PLoS One* 14:e0222190. doi:10.1371/journal.pone.0222190

Bullimore SR, Leonard TR, Rassier DE, Herzog W. 2007. History-dependence of isometric muscle force: Effect of prior stretch or shortening amplitude. *Journal of Biomechanics* 40:1518–1524. doi:10.1016/j.jbiomech.2006.06.014

Cook CS, McDonagh MJ. 1995. Force responses to controlled stretches of electrically stimulated human muscle-tendon complex. *Experimental Physiology* 80:477–490.

Coombes BK, Tucker K, Vicenzino B, Vuvan V, Mellor R, Heales L, Nordez A, Hug F. 2018. Achilles and patellar tendinopathy display opposite changes in elastic properties: A shear wave elastography study. *Scand J Med Sci Sports* 28:1201–1208. doi:10.1111/sms.12986

Draganich LF, Andriacchi TP, Andersson GB. 1987. Interaction between intrinsic knee mechanics and the knee extensor mechanism. *J Orthop Res* 5:539–547. doi:10.1002/jor.1100050409

Edman KA, Elzinga G, Noble MI. 1982. Residual force enhancement after stretch of contracting frog single muscle fibers. *The Journal of General Physiology* 80:769–784. doi:10.1085/jgp.80.5.769

Edman KA, Elzinga G, Noble MI. 1978. Enhancement of mechanical performance by stretch during tetanic contractions of vertebrate skeletal muscle fibres. *The Journal of Physiology* 281:139–155.

Groeber M, Stafilidis S, Seiberl W, Baca A. 2020. Contribution of Stretch-Induced Force Enhancement to Increased Performance in Maximal Voluntary and Submaximal Artificially Activated Stretch-Shortening Muscle Action. *Frontiers in Physiology* 11.

Hisey B, Leonard TR, Herzog W. 2009. Does residual force enhancement increase with increasing stretch magnitudes? *Journal of Biomechanics* 42:1488–1492. doi:10.1016/j.jbiomech.2009.03.046

Martin JA, Brandon SCE, Keuler EM, Hermus JR, Ehlers AC, Segalman DJ, Allen MS, Thelen DG. 2018. Gauging force by tapping tendons. *Nature Communications* 9:1592. doi:10.1038/s41467-018-03797-6

Mogi Y. 2020. The effects of growth on structural properties of the Achilles and Patellar tendons: A cross-sectional study. *Physiol Rep* 8:e14544. doi:10.14814/phy2.14544

Power GA, Makrakos DP, Rice CL, Vandervoort AA. 2013. Enhanced force production in old age is not a far stretch: An investigation of residual force enhancement and muscle architecture. *Physiological Reports* 1:e00004. doi:10.1002/phy2.4

Shim J, Garner B. 2012. Residual force enhancement during voluntary contractions of knee extensors and flexors at short and long muscle lengths. *Journal of Biomechanics* 45:913–918. doi:10.1016/j.jbiomech.2012.01.026

Tsaopoulos DE, Maganaris CN, Baltzopoulos V. 2007. Can the patellar tendon moment arm be predicted from anthropometric measurements? *Journal of Biomechanics* 40:645–651. doi:10.1016/j.jbiomech.2006.01.022